# Molecular basis for PrimPol recruitment to replication forks by RPA

Thomas A. Guilliam[1,*], Nigel C. Brissett[1,*], Aaron Ehlinger[2,*], Benjamin A. Keen[1], Peter Kolesar[1], Elaine M. Taylor[3], Laura J. Bailey[1], Howard D. Lindsay[3], Walter J. Chazin[2] & Aidan J. Doherty[1]

DNA damage and secondary structures can stall the replication machinery. Cells possess numerous tolerance mechanisms to complete genome duplication in the presence of such impediments. In addition to translesion synthesis (TLS) polymerases, most eukaryotic cells contain a multifunctional replicative enzyme called primase–polymerase (PrimPol) that is capable of directly bypassing DNA damage by TLS, as well as repriming replication downstream of impediments. Here, we report that PrimPol is recruited to reprime through its interaction with RPA. Using biophysical and crystallographic approaches, we identify that PrimPol possesses two RPA-binding motifs and ascertained the key residues required for these interactions. We demonstrate that one of these motifs is critical for PrimPol's recruitment to stalled replication forks in vivo. In addition, biochemical analysis reveals that RPA serves to stimulate the primase activity of PrimPol. Together, these findings provide significant molecular insights into PrimPol's mode of recruitment to stalled forks to facilitate repriming and restart.

[1] Genome Damage and Stability Centre, School of Life Sciences, University of Sussex, Brighton BN1 9RQ, UK. [2] Departments of Biochemistry and Chemistry and Center for Structural Biology, Vanderbilt University School of Medicine, Nashville, Tennessee 37232, USA. [3] Lancaster Medical School, Faculty of Health and Medicine, Lancaster University, Lancaster LA1 4YQ, UK. * These authors contributed equally to this work. Correspondence and requests for materials should be addressed to A.J.D. (email: ajd21@sussex.ac.uk).

An intricate complex of molecular machines, known collectively as the replisome, duplicate the genome during DNA replication. At the heart of the replisome are the replicative polymerases, which synthesize DNA with a high degree of accuracy and efficiency. Nevertheless, these enzymes are vulnerable to aberrations in the template strand, including DNA lesions and secondary structures, which can lead to replication stalling at these sites. A number of mechanisms exist to permit the resumption of replication during these events[1–3]. One such mechanism is the generation of a nascent primer downstream of the obstacle, termed repriming[4]. This allows the replisome to effectively skip over the impediment and restart replication.

Although Pol α-primase was thought to be the only eukaryotic primase, we now know eukaryotes possess a second primase known as primase–polymerase (PrimPol)[5–7]. PrimPol is a member of the archaeo-eukaryotic primase (AEP) superfamily, whose members fulfil a range of roles in DNA replication, repair and damage tolerance[8], and it possesses both primase and TLS polymerase activities[5,6]. Evidence is accumulating that suggests the primary role of PrimPol in vivo is to reprime DNA replication downstream of DNA damage lesions and secondary structures[9–12]. Despite assisting the replisome through this role, PrimPol could be potentially deleterious to genomic integrity due to its low fidelity and penchant for generating frame-shift mutations[13]. As a result, the enzyme must be tightly regulated and only allowed to contribute to DNA synthesis when absolutely required.

We previously identified the nuclear and mitochondrial (mt) single-stranded DNA-binding (SSB) proteins, replication protein A (RPA) and mtSSB, as PrimPol-interacting partners in vivo. Using biochemical and biophysical approaches, we demonstrated that PrimPol interacts with the RPA70N subunit of RPA and that both of its single-stranded DNA-binding proteins binding partners serve to restrict the contribution of PrimPol to DNA synthesis during replication, thereby limiting the opportunity for mutagenesis[13]. PrimPol was also identified as an RPA-binding partner by another group[7] who suggested that RPA may act to recruit PrimPol to stalled replication forks in vivo.

In this study, we present an in-depth interrogation of the interaction between PrimPol and RPA, identifying that PrimPol possesses two RPA-binding motifs (RBMs, RBM-A and RBM-B) in its C-terminal domain (CTD). Both of these motifs are able to bind directly to RPA70N, a primary recruitment domain of RPA that mediates interactions with a number of DNA damage response proteins, including p53, ATRIP, RAD9 and MRE11 (ref. 14). Using biophysical and crystallographic approaches, we elucidated the molecular basis of each of the PrimPol-RBM interactions and identified the critical residues involved in each complex. We generated PrimPol RBM mutants in vivo and analysed the importance of each of these sites for PrimPol's role in DNA damage tolerance. We identify that RBM-A is the primary mediator of PrimPol's interaction with RPA in vivo, with RBM-B potentially playing a more secondary role. The interaction between RBM-A and RPA70N is critical for the recruitment of PrimPol to chromatin and for stimulating the enzyme's role in repriming DNA replication. Notably, mutations in both RBMs affecting key residues involved in binding (for example, F522V and I554T) have been identified in cancer patient cell lines and these mutations are sufficient to abrogate binding of RPA70N to the affected RBM. Collectively, these results describe the molecular and cellular basis for PrimPol's recruitment by RPA to stalled replication forks and demonstrates the importance of these interactions for maintaining PrimPol's functions in replication fork progression in vivo.

## Results

**PrimPol's CTD interacts with RPA70N.** Previously, we identified that full-length human PrimPol interacts directly with the RPA70N domain of RPA70 and deletion of the C-terminal RPA-binding domain (RBD; amino acids 480–560) ablated this interaction[13]. To determine if PrimPol's RBD (480–560) is sufficient to mediate binding, we performed analytical gel filtration chromatography (GFC) on human PrimPol$_{RBD}$ titrated with RPA70N (Supplementary Fig. 1a). With one equivalent of RPA70N added, a bimodal peak appears with broadened densities between a position near free PrimPol$_{RBD}$ and a peak presumably of the complex (blue dot trace). With two equivalents of RPA70N added, the peak at the PrimPol$_{RBD}$ position is much weaker, while the complex elutes slightly earlier and increases in intensity (blue dash trace). With four equivalents of RPA70N added, the complex peak increases in intensity, the free PrimPol$_{RBD}$ peak disappears and a peak at the free RPA70N position becomes visible (blue solid trace). This data indicates a heterogeneous interaction, most likely from two binding sites of similar affinity (Supplementary Fig. 1a). The stoichiometry of the binding is most likely 2:1 RPA70N:PrimPol$_{RBD}$ due to the complete disappearance of the individual RPA70N peak at this ratio (Supplementary Fig. 1a). This stoichiometry was further confirmed by multiangle light scattering (MALS) analysis of the eluted peak fractions, identifying a heterogenous mix of both 1:1 and 2:1 RPA70N:PrimPol$_{RBD}$ complexes (Supplementary Fig. 1b). PrimPol$_{RBD}$ had a much lower retention volume (10.39 ml) than expected for an 8.8 kDa protein, corresponding to a predicted molecular weight of ~42 kDa if the protein was globular. Nevertheless, circular dichroism (CD) and dynamic light scattering revealed that PrimPol$_{RBD}$ is monomeric in solution with a largely non-globular structure (Supplementary Fig. 1c and d).

NMR spectroscopy was next utilized to cross-validate this interaction. To this end, $^{15}$N-enriched PrimPol$_{RBD}$ was produced and analysed by two-dimensional (2D) $^{15}$N-$^{1}$H heteronuclear single-quantum coherence (HSQC) NMR (Supplementary Fig. 1e). The low dispersion observed in the $^{1}$H dimension of the spectrum is characteristic of a protein with non-globular structure. Upon addition of unlabelled RPA70N to a twofold molar excess, there was a significant effect on the spectrum, with peaks attenuating, broadening or shifting. These observations confirm that there is an interaction between the two proteins. We also observed significant peak shifting and disappearance in the corresponding spectrum of $^{15}$N-enriched RPA70N in the presence of a twofold excess of unlabelled PrimPol$_{RBD}$ (Supplementary Fig. 1f). The large number of peaks affected and the variety of effects on the signals suggest the interaction is not mediated by a single high-affinity site, but rather some form of heterogeneous binding.

**PrimPol RBD contains a conserved RPA-binding motif.** RPA70N contains a prominent surface cleft that binds many interacting partners, including RAD9, MRE11, ATRIP and p53 (ref. 14). These partners utilize a similar highly negatively charged motif, which interacts with the exposed basic residues in the RPA70N cleft[14]. Examination of the human PrimPol sequence revealed a divergent acidic motif within its RBD (residues 513–527; Fig. 1a), we termed this motif RBM-A.

To investigate the potential interaction between PrimPol's RBM-A and RPA70N, we employed NMR spectroscopy using an RBM-A peptide (RBM-A$_{510–528}$). An overlay of 2D $^{15}$N-$^{1}$H HSQC spectra of $^{15}$N-enriched RPA70N acquired in the absence (black) and presence (red) of a twofold excess of RBM-A$_{510–528}$ revealed significant chemical shift perturbations (CSPs) induced

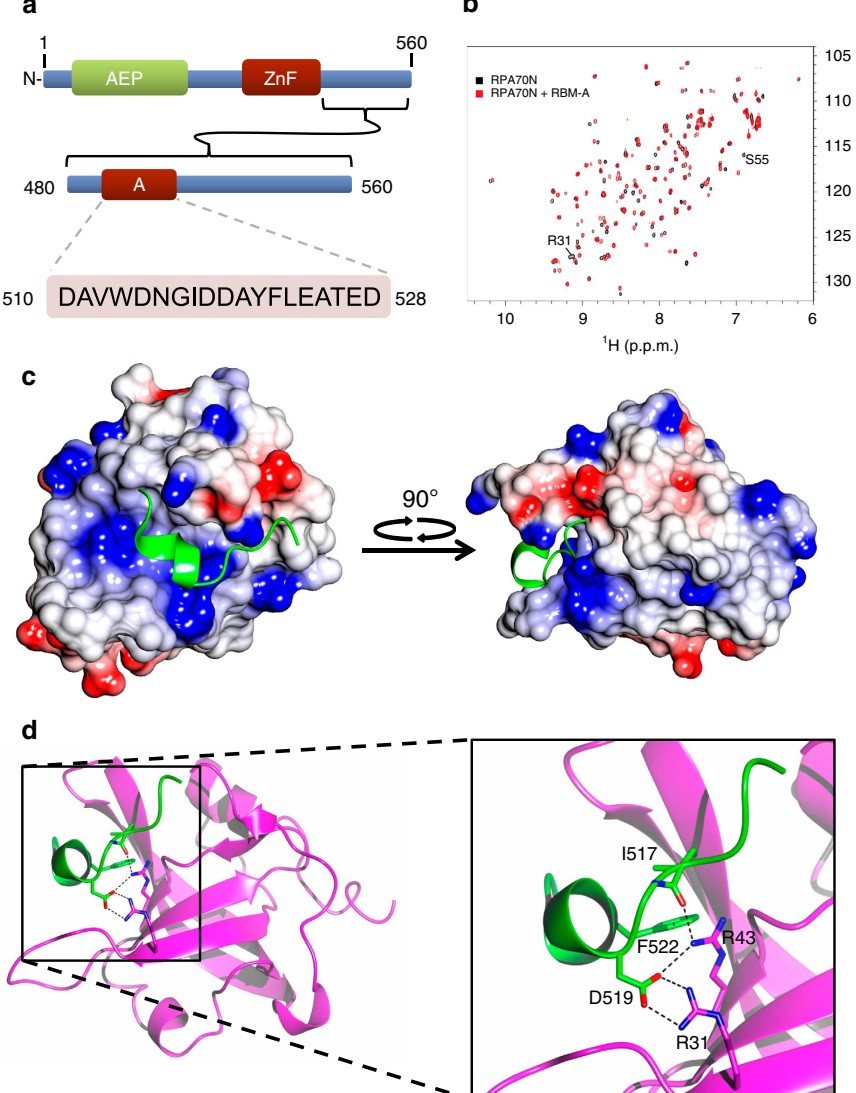

**Figure 1 | PrimPol possesses a conserved RBM that binds to the basic cleft of RPA70N.** (**a**) Schematic showing the sequence of PrimPol's RBM-A (residues 510–528), located in the C-terminal RBD (residues 480–560). (**b**) $^{15}$N-$^1$H HSQC spectra showing RPA70N in the absence (black) or presence (red) of twofold molar excess of unlabelled RBM-A peptide. (**c**) Electrostatic surface model of RPA70N with RBM-A (green) bound in the basic cleft. Basic and acidic surfaces are coloured blue and red, respectively. (**d**) Key stabilizing interactions of RBM-A (green) in the RPA70N basic cleft (purple). RBM-A binds between β sheets in the β barrel of RPA70N. Of particular importance for binding are the electrostatic interactions of D519 with the side chains of two arginines (R31 and R43) in the RPA70N basic cleft.

by binding of the peptide (Fig. 1b). The CSPs above a defined threshold ($\Delta\delta > 0.1$) were mapped onto the RPA70N structure and compared with the corresponding CSPs caused by the binding of other RPA-interacting proteins; ATRIP, Rad9 and MRE11 (Supplementary Fig. 2a and b)[14]. Similar to these binding partners, RBM-A bound within the basic cleft of RPA70N. Together, these studies establish that RBM-A interacts with RPA via the basic cleft of RPA70N.

**Molecular basis for RBM-A RPA70N interaction**. To determine the molecular basis for RPA70N binding to the RBM-A site of PrimPol, RPA70N$^{E7R}$ (an RPA70N mutant optimized for crystallization of complexes[15]) and the RBM-A peptide residues (PrimPol$_{514–528}$) were co-crystallized. Co-crystals contained a 1:1 molar ratio in a $P2_12_12_1$ crystal lattice (Fig. 1c,d). The statistics for data processing are summarized in Table 1. Continuous electron density covers the entirety of RPA70N$^{E7R}$ and 12 residues

(514–525) of the 15-mer PrimPol$_{514–528}$ peptide are visible in the electron density maps. Within this short peptide, residues aspartate 519 to leucine 523 are α-helical in content. Given that no α-helices were identified from circular dichroism of the free RBD, it is likely that the α-helical peptide identified here is induced upon binding. A striking feature of this α-helix is that the primary interactions with the basic cleft of RPA70N$^{E7R}$ are via salt bridges between aspartate 519 of PrimPol and RPA70N$^{E7R}$ arginines R31 and R43. Hydrogen bonds are also found between isoleucine 517 of PrimPol and RPA70N$^{E7R}$ arginine 43. In addition to the ionic interactions, PrimPol phenylalnnine 522 sits in a hydrophobic pocket made up of RPA70N$^{E7R}$ serine 55, methionine 57 and valine 93. Isoleucine 517 of PrimPol also has an aliphatic interaction with the side chain of RPA70N$^{E7R}$ arginine 91 (Fig. 1c,d). The combination of these electrostatic and hydrophobic interactions drives the stabilization of this complex.

| | RPA70N$^{E7R}$/ PrimPol$_{514-528}$ | RPA70N$^{E7R}$/ PrimPol$_{480-560}$ |
|---|---|---|
| **Table 1 \| Data collection and refinement statistics (molecular replacement).** | | |
| *Data collection* | | |
| Space group | $P2_12_12_1$ | $P2_12_12_1$ |
| Cell dimensions | | |
| $a, b, c$ (Å) | 37.86, 53.09, 54.63 | 38.05, 53.49, 53.9 |
| $\alpha, \beta, \gamma$ (°) | 90.00, 90.00, 90.00 | 90.00, 90.00, 90.00 |
| Resolution (Å) | 31.12 (2.00)* | 16.25 (1.28)* |
| $R_{sym}$ or $R_{merge}$ | 0.217 (0.751) | 0.044 (0.655) |
| $I/\sigma I$ | 10.9 (3.0) | 26.9 (2.7) |
| Completeness (%) | 99.6 (98.1) | 99.8 (98.2) |
| Redundancy | 12.8 (10.4) | 12.1 (7.4) |
| | | |
| *Refinement* | | |
| Resolution (Å) | 31.12 (2.00) | 16.25 (1.28) |
| No. of reflections | 7,825 | 28,966 |
| $R_{work}/R_{free}$ | 0.1873/0.2286 | 0.1537/0.1785 |
| No. of atoms | 1,148 | 1,210 |
| Protein | 1,074 | 1,070 |
| Ligand/ion | | |
| Water | 74 | 140 |
| *B*-factors | | |
| Protein | 25.25 | 19.73 |
| Ligand/ion | | |
| Water | 31.23 | 32.95 |
| R.m.s.d. | | |
| Bond lengths (Å) | 0.004 | 0.007 |
| Bond angles (°) | 0.785 | 0.912 |

r.m.s.d., root mean squared deviation.
Data from one crystal for each structure.
*Values in parentheses are for highest-resolution shell.

**PrimPol RBD contains a second RPA-binding motif.** To determine the molecular basis for binding of PrimPol RBM-A to RPA70N, in the wider context of the whole RBD, we co-crystallized a complex of PrimPol$_{480-560}$ bound to RPA70N$^{E7R}$. Again, co-crystals contained a 1:1 molar ratio in an orthorhombic $P2_12_12_1$ crystal lattice (Fig. 2b,c,e). The statistics for data processing are summarized in Table 1. Similar to the RBM-A peptide, continuous electron density covers the entirety of RPA70N$^{E7R}$ and nine amino acids of an α-helical peptide from PrimPol$_{480-560}$ are visible in the electron density maps. Surprisingly, model building and density refinement revealed that RPA70N bound to PrimPol residues 546–560 (Fig. 2a,b) rather than RMB-A. An excellent fit to the high-resolution (1.28 Å) electron density is evident for amino acids 548–556, despite residues 480–547 not being visible in the area of contiguous electron density (Fig. 2b). As PrimPol's RBD lacks significant secondary structure or globular fold, and residues 480–547 are not tethered to RPA in the lattice, we expect that these residues remain flexible in the crystal and this disorder inhibits their resolution.

The crystal structure revealed that the second RBM, termed RBM-B, also binds to the basic cleft of RPA70N (Fig. 2c). Like RBM-A, RBM-B has a low pI (pI = 3.25) but this motif contains two adjacent Asp-Glu motifs instead of the typical di-Asp motif (Fig. 2a), not previously identified in the RPA70N binding motifs of other RPA partner proteins. To confirm that the interaction observed in the crystal is a *bona fide* RPA70N binding motif, we examined the binding to RPA70N of a PrimPol$_{542-560}$ peptide using $^{15}$N-$^1$H HSQC NMR. The spectrum of $^{15}$N-enriched RPA70N in the absence and presence of a twofold molar excess of the RBM-B peptide reveals significant CSPs induced by the binding of PrimPol RBM-B (Fig. 2d). As observed for the RBM-A

titration, the RBM-B peptide causes CSPs of residues in RPA70N's basic cleft, including characteristic residues S55 and R31 (Fig. 2d, Supplementary Fig. 4a). Together, these data demonstrate that PrimPol's RBD contains a second independent RPA70N binding motif.

**Molecular basis for RBM-B RPA70N interaction.** Notably, the RBM-A sequence and the structure of its complex with RPA70N is at odds with the well defined 'canonical' RBMs (for example, p53, ATRIP) and likewise, the structure of RBM-B bound to RPA70N in the crystal of PrimPol RBD confirms these distinct features (Figs 1d and 2e, Supplementary Fig. 3a). Notably, these differences arose despite the absence of any significant effects on the structure of RPA70N. The orientation of the RBM-B helix is stabilized by a number of electrostatic interactions (Fig. 2e). The aspartate at position 551 of PrimPol is perhaps the most important point of contact as it interacts with the two arginines of RPA70N (R31 and R43) and a threonine (T34) side chain, as well as the backbone amide N-H of T34. The carbonyl group of PrimPol's isoleucine at position 549 likely acts as a hydrogen bond acceptor for the RPA's R43. The glutamate at position 548 forms an electrostatic interaction with an arginine (R91) on the other side of RPA70N's β-barrel, acting to secure the helix of PrimPol in this orientation. These electrostatic interactions are of paramount importance in the binding of PrimPol's RBM-B to RPA70N *in vitro* (Fig. 2b,e).

Comparison of the RBM-A and RBM-B structures reveals that the peptides adopt almost identical helical conformations that occupy the basic cleft in a similar fashion (Supplementary Fig. 3a–c). Intriguingly, the interactions between PrimPol's RBM-A/B and RPA70N are significantly different from the interactions reported for either a modified ATRIP stapled peptide or a p53 peptide bound to RPA70N (refs 16,17). A superposition of the modified ATRIP peptide with RBMs shows that the two helices bind in a similar region to RPA70N however, they are in opposite orientations (Supplementary Fig. 3a–f). In addition, the main interaction of the modified ATRIP peptide is of its modified 3,4-dichlorophenyl amino acid into a hydrophobic pocket on RPA70N, and in p53 there is a phenylalanine residue that extends into this pocket. This pocket is also the region where a RPA70N binding inhibitor (VUO79104) bound to a co-crystal structure[15]. PrimPol's RBM-A and RBM-B have hydrophobic residues phenylalanine (F522) and isoleucine (I554) that occupy the hydrophobic pocket on RPA70N (Figs 1d and 2e). F522 forms hydrophobic non-bonding contacts with a serine (S55) methionine (M57) and a valine (V93) of RPA70N in this pocket. Whereas, I554 forms hydrophobic non-bonding contacts with the methionine and valine only. We propose that the RPA70N binding modes observed for PrimPol may be more 'physiological' as the bound motifs are not modified in any way, unlike p53 and ATRIP where co-crystals could only be obtained by altering the peptides[16].

**Exchangeable binding of PrimPol RBMs to RPA70N.** As both RBM-A and RBM-B interact in the basic cleft, we next analysed whether these sites bind coordinately or competitively. To this end, we constructed RBM-A (D514R/D518R/D519R) and B (480–546 truncation) knockout (KO) mutants in the PrimPol$_{RBD}$ construct. Both NMR and GFC were used to analyse the binding of these mutants to RPA70N. Similar to results observed with PrimPol$_{RBD}$, PrimPol$_{A-KO}$ and RPA70N eluted together as a well defined multimeric complex from GFC (Fig. 3a). In addition, HSQC titrations of twofold molar addition of PrimPol$_{A-KO}$ into $^{15}$N-enriched RPA70N produced clear evidence of binding (Fig. 3b). Likewise, PrimPol$_{B-KO}$ was found to bind RPA70N in

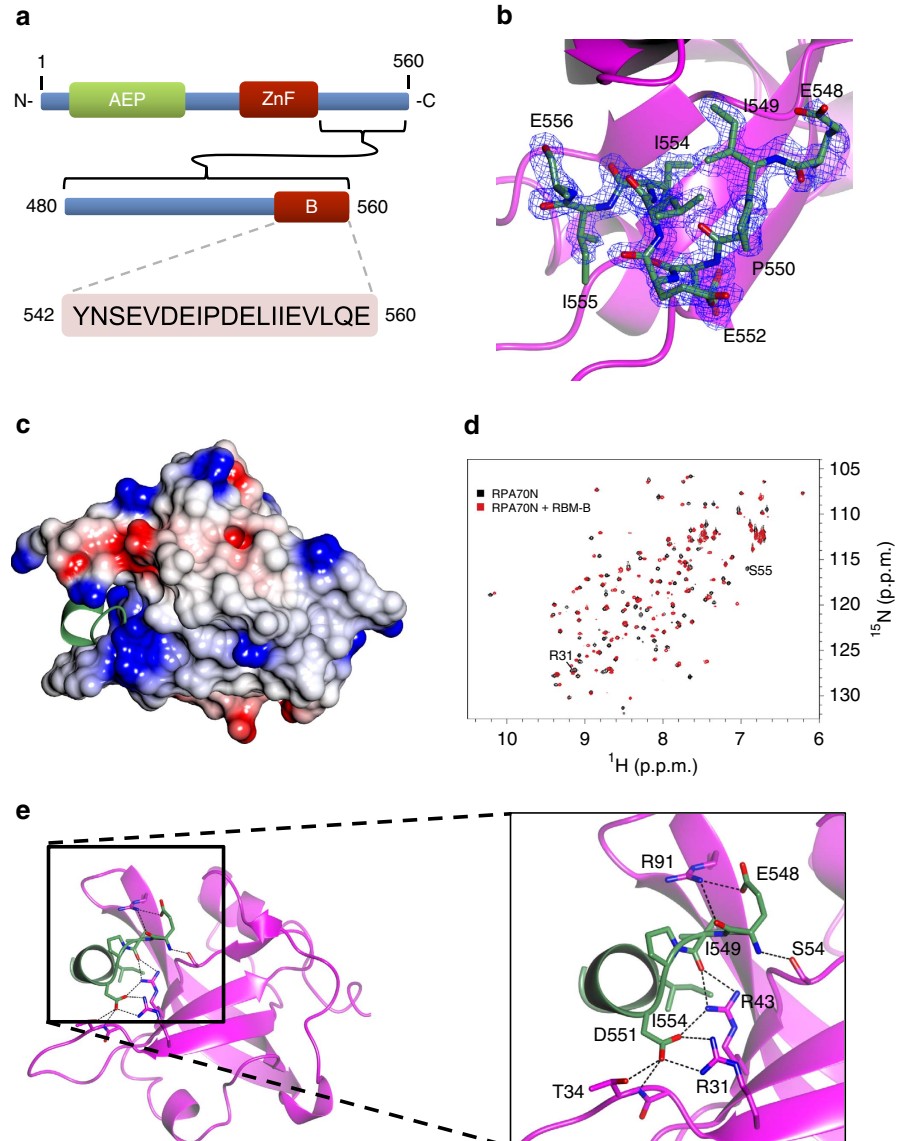

**Figure 2 | PrimPol possesses a second RBM that also binds to the basic cleft of RPA70N.** (**a**) The sequence of PrimPol's RBM-B (residues 542–560), located in the C-terminal RBD (residues 480–560). (**b**) The continuous electron density of RBM-B residues 548–556 in the complex with RPA70N. (**c**) Electrostatic surface model of RPA70N with RBM-B (green) bound in the basic cleft. Basic and acidic surfaces are coloured blue and red, respectively. (**d**) $^{15}N$-$^{1}H$ HSQC spectra showing RPA70N in the absence (black) or presence (red) of a twofold molar excess of unlabelled RBM-B peptide. (**e**) Key stabilizing interactions of RBM-B (green) in the RPA70N basic cleft (purple). RBM-B binds between β sheets in the β barrel of RPA70N. D551 is of particular importance as it forms a number of electrostatic interactions with both the side chains and a backbone amide NH of the RPA70N peptide.

both GFC and NMR analyses (Fig. 3c,d). Notably, in each GFC analysis a small fraction of unbound RPA70N was observed, unlike GFC using the wild-type RBD. Similarly, NMR analysis produced results mimicking those of the isolated motifs, suggesting that both mutants retain binding activity characteristic of the unaltered RBM-A and B. By overlaying the HSQC spectra of RPA70N in the presence of WT, or mutant RBM-A/B, RBD, we identified that, while most of the signals from the complex with mutant RBM-A/B RBD are identical to the complex with WT RBD (Supplementary Fig. 4a), some peaks from the RBM-A/B-bound spectra do not overlap. These signals correspond to residues that attenuate or disappear in the complex with WT RBD. Analysis of this phenomenon suggests that RPA70N binds to both sites in solution and this process is exchangeable. This is consistent with ITC data showing that PrimPol$_{A-KO}$ and PrimPol$_{B-KO}$ bind to RPA70N with statistically identical affinities

of $7.8 \pm 0.6\,\mu M$ and $6.7 \pm 1.5\,\mu M$, respectively (Supplementary Fig. 4a and b).

In contrast, there was no observed binding in the GFC or NMR when the 'double' mutant (PrimPol$_{A/B-KO}$) was incubated with RPA70N (Fig. 3e,f). In addition, no heat of binding was observed by ITC (Supplementary Fig. 4c). Therefore, while retaining either one of these domains is sufficient to maintain RPA70N binding *in vitro*, knocking out both RBM-A and RBM-B completely abrogates binding. This indicates that there are no additional RPA70N binding sites beyond RBM-A and RBM-B.

To obtain less perturbing mutants for experiments *in vivo*, we analysed 'finer' point mutants of both RBM-A and B, based on the crystallographic data. We found that the PrimPol$_{A-RA}$ (D519R/F522A) and PrimPol$_{B-RA}$ (D551R/I554A) mutants retained the ability to bind RPA70N in GFC. However, binding was lost when all four residues were mutated (Supplementary

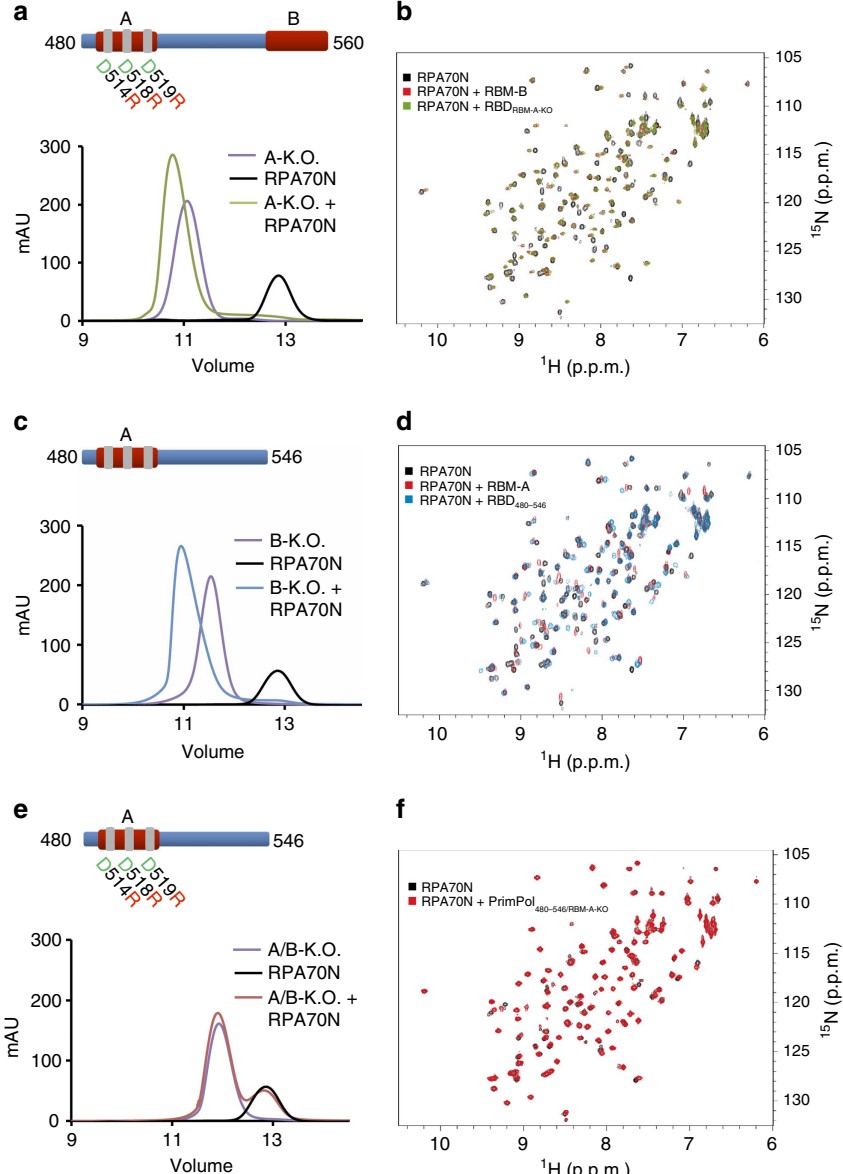

**Figure 3 | RPA70N dynamically interacts with both RBM-A and RBM-B.** (**a**) Mutation of RBM-A does not abolish binding of PrimPol's RBD to RPA70N. Chromatographs showing the retention volumes of RBD$_{A-KO}$ (purple), RPA70N (black) and RBD$_{A-KO}$ with RPA70N in a 1:1 ratio (green). (**b**) $^{15}$N-$^{1}$H HSQC spectra showing RPA70N alone (black), in the presence of twofold molar excess of either RBD$_{A-KO}$ (green) or RBM-B peptide (residues 542–560) (red). The perturbations observed for RBD$_{A-KO}$ are similar to those induced by the RBM-B peptide. (**c**) Truncation of RBM-B does not prevent binding of PrimPol's RBD to RPA70N. Chromatographs showing the retention volumes of RBD$_{B-KO}$ (purple), RPA70N (black) and RBD$_{B-KO}$ with RPA70N in a 1:1 ratio (blue). (**d**) $^{15}$N-$^{1}$H HSQC spectra showing RPA70N alone (black) or in the presence of twofold molar excess of RBD$_{B-KO}$ (blue) or RBM-A peptide (residues 510–528) (red). The perturbations observed for RBD$_{B-KO}$ are similar to those induced by the RBM-A peptide. (**e**) Mutation of both RBM-A and RBM-B abolishes the binding of PrimPol's RBD to RPA70N. Chromatographs showing the retention volumes of RBD$_{A/B-KO}$ (purple), RPA70N (black) and RBD$_{A/B-KO}$ with RPA70N in a 1:1 ratio (red). (**f**) $^{15}$N-$^{1}$H HSQC spectra showing RPA70N alone (black) or in the presence of twofold molar excess of RBD$_{A/B-KO}$ (red). The near identity of the two spectra indicates there is no interaction.

Fig. 5a). We additionally analysed these mutations in the context of the full-length protein and RBD (480–560) using the yeast two-hybrid assay. Here, PrimPol$_{A-RA}$ and PrimPol$_{B-RA}$ exhibited decreased binding to RPA70N, with an additional decrease when both sites were mutated. Near identical results were observed when analysing both the full-length enzyme and RBD, confirming that both RBM-A and RBM-B are able to bind RPA70N when outside their innate vertebrate cell environment (Supplementary Fig. 5b). These results, therefore, confirmed that each RBM is accessible for RPA70N binding in the context of the full-length protein. Furthermore, they provided minimally perturbing

PrimPol variants to probe the functional significance of the RPA interaction, and the contributions of the two RBMs, *in vivo*.

**RBM-A mediates the PrimPol-RPA interaction *in vivo*.** To ascertain the importance of each RBM in mediating PrimPol's interaction with RPA *in vivo*, we introduced doxycycline-inducible N-terminal FLAG-tagged PrimPol variants lacking either RBM (A or B), or both, into HEK-293 derivative cells (Flp-In T-Rex-293) (Fig. 4a,b) and performed co-immunoprecipitation experiments. We found that RPA co-precipitates with

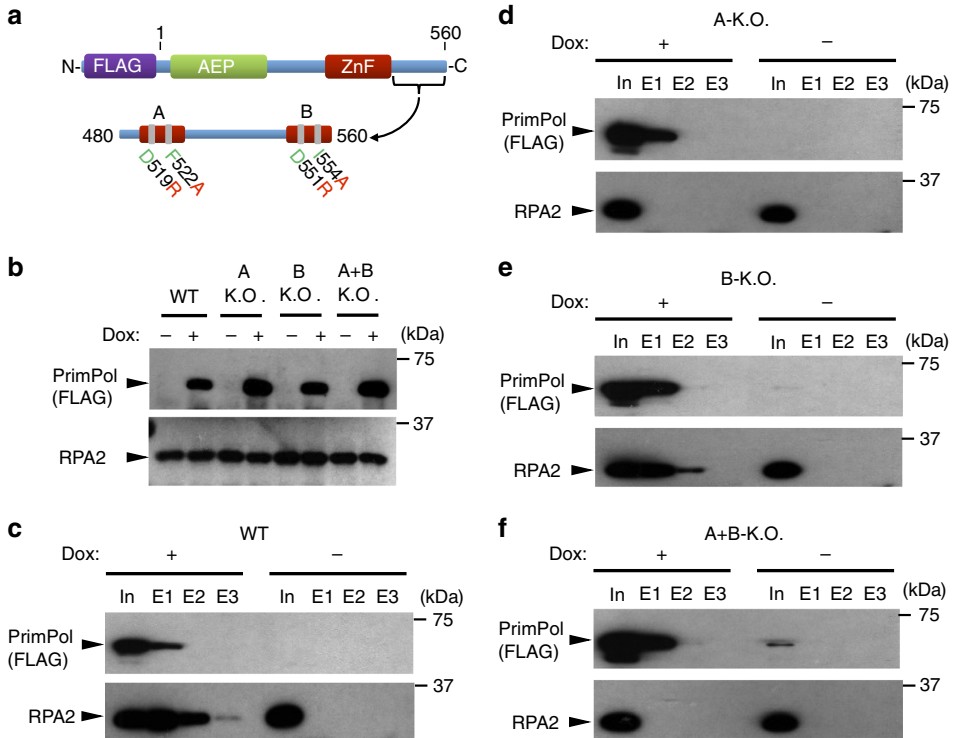

**Figure 4 | PrimPol RBM-A is critical for RPA-binding in vivo.** (**a**) Schematic detailing the domain architecture of N-terminal FLAG-tagged PrimPol transfected into HEK-293 derivative cells (Flp-In T-Rex-293). The RBD (480–560) containing the RBM-A and B sites is shown below with the mutations forming the A-KO (D519R and F522A) and B-KO (D551R and I554A) highlighted. (**b**) Flp-In T-Rex-293 cells were transfected with wild-type and RBM-A and B mutated PrimPol. Expression was confirmed by addition of 10 ng ml$^{-1}$ doxycycline (indicated by Dox ± on figure) for 24 h and subsequent western blotting. (**c**) Flp-In T-Rex-293 cells transfected with FLAG-tagged wild-type (WT) PrimPol were grown in the presence or absence of doxycycline (10 ng ml$^{-1}$, 24 h), FLAG-PrimPol was immunoprecipitated from the soluble cell lysate using anti-FLAG antibody and western blotted for PrimPol (anti-FLAG) and RPA (anti-RPA2). The presence and absence of doxycycline is indicated by ± Dox, 'In' indicates the input, 'E1', 'E2' and 'E3', indicate elutions 1, 2 and 3, respectively. (**d**) Immunoprecipitation of FLAG-PrimPol$_{A-KO}$ (D519R/F522A) from Flp-In T-Rex-293 cells grown in the presence and absence of doxycycline. (**e**) Immunoprecipitation of FLAG-PrimPol$_{B-KO}$ (D551R/I554A) from Flp-In T-Rex-293 cells grown in the presence and absence of doxycycline. (**f**) Immunoprecipitation of FLAG-PrimPol$_{A+B-KO}$ (D519R/F522A and D551R/I554A) from Flp-In T-Rex-293 cells grown in the presence and absence of doxycycline.

FLAG-PrimPol *in vivo* when both RBMs are unmodified (Fig. 4c), confirming that FLAG-PrimPol interacts with RPA in a damage-independent manner, as observed previously[7,13]. In addition, FLAG-PrimPol$_{RBD}$ (the CTD only) also co-precipitated with RPA, supporting our *in vitro* data and previous reports that PrimPol interacts with RPA via its CTD (Supplementary Fig. 5c)[7,13]. Interestingly, we observed that mutation of RBM-A (D519R/F522A) alone abolishes this interaction, despite the protein possessing an intact RBM-B (Fig. 4d). Furthermore, when RBM-B is mutated (D551R/I554A), but RBM-A was intact, a reduced, but significant, amount of RPA still co-precipitated with FLAG-PrimPol (Fig. 4e). Unsurprisingly, when both RBMs were mutated, the interaction with RPA was again lost (Fig. 4f). Together, these findings identify that RBM-A is the primary mediator of PrimPol's interaction with RPA *in vivo* and residues D519 and F522 as essential for forming the complex. In contrast, RBM-B appears to play a more secondary role in RPA-binding *in vivo*.

**PrimPol requires an RPA interaction to function *in vivo*.** PrimPol has previously been shown to promote DNA replication fork restart following ultraviolet damage by repriming[5,9,10]. To define the importance of each RBM on PrimPol's role during this process, we complemented PrimPol$^{-/-}$ DT40 cells with RBM-A (D519R/F522A) and RBM-B (D551R/I554A) mutants (Fig. 5a)

and performed DNA fibre analysis on these cells in the presence of ultraviolet damage. We labelled replicating cells with the nucleotide analogue chlorodeoxyuridine (CldU) for 20 min, cells were then ultraviolet-C irradiated (20 J m$^{-2}$) and labelled with a second nucleotide analogue, iododeoxyuridine (IdU), for an additional 20 min (Fig. 5b). Following detection by immunofluorescence, the degree of fork stalling after ultraviolet damage in the PrimPol RBM-mutant cells was determined by analysing the CldU:IdU tract length ratios. An increase in this ratio indicates a shorter IdU tract and therefore an increase in the amount of fork stalling or slowing following ultraviolet-C irradiation.

Cells expressing RBM-A-mutant PrimPol presented a significant increase in the mean CldU:IdU tract length ratio when compared to cells complemented with wild-type PrimPol (Fig. 5c,d). In addition, these cells displayed more variation in CldU:IdU ratios with an increase in the percentage of forks with higher ratios (Fig. 5c,d). This indicates that there was an increase in fork stalling events, or a decreased ability to restart stalled forks, in these cells. The observed effect was not as severe as that seen in PrimPol$^{-/-}$ cells, however given that RBM-A-mutant PrimPol is catalytically identical to wild-type PrimPol, and over-expressed in these cells, this was not surprising. This result suggests that mutation of RBM-A affects PrimPol's recruitment to stalled replication forks and therefore causes an impairment in the ability to restart these forks. Given the level of over-expression

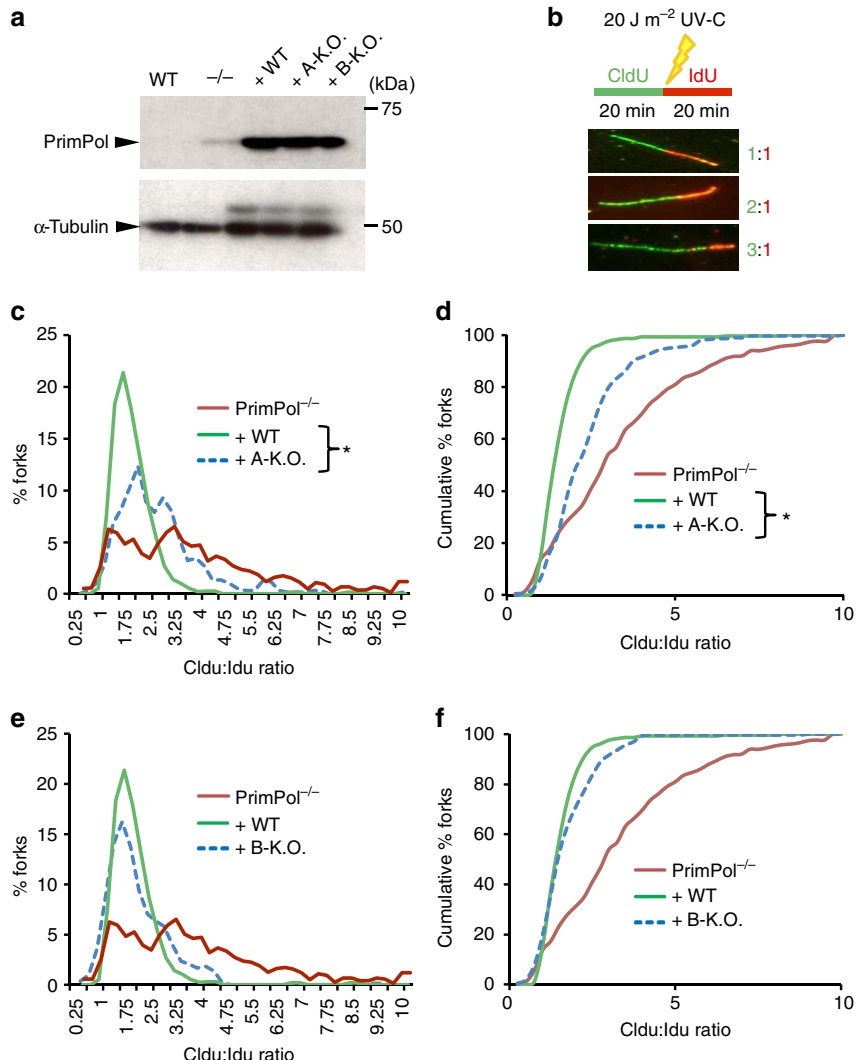

**Figure 5 | RBM-A is required for PrimPol function in DNA replication restart.** (**a**) PrimPol$^{-/-}$ DT40 cells were complemented with un-tagged human PrimPol constructs; wild-type hPrimPol ( + WT), hPrimPol$_{D519R/F522A}$ ( + A-KO) and hPrimPol$_{D551R/I554A}$ ( + B-KO). 'WT' indicates lysate from wild-type DT40 cells, ' − / − ' indicates lysate from PrimPol$^{-/-}$ DT40 cells. (**b**) DNA fibre analysis was performed on DT40 cells expressing each PrimPol construct. Cells were ultraviolet-C irradiated (20 J m$^{-2}$) between the CldU and IdU labelling periods (each 20 min). Representative DNA fibres showing 1:1, 2:1 and 3:1 CldU:IdU ratios are presented; >100 individual DNA fibres were scored for each experiment. (**c**) Mutation of RBM-A causes increased fork stalling following ultraviolet-C irradiation. Data are representative of the means of three individual experiments and were subject to an unpaired *t*-test showing a significant difference between the mean CldU/IdU ratio for the ' + WT hPrimPol' and ' + A-KO hPrimPol' data sets ($P < 0.05$). (**d**) DNA fibre analysis from the ' + A-KO hPrimPol' DT40 cells presented as a cumulative percentage of forks at each ratio. (**e**) Mutation of RBM-B does not significantly alter the level of fork stalling following ultraviolet-C irradiation. DNA fibre analysis of the ' + B-KO hPrimPol' DT40 cells, showing the percentage of forks at each CldU:IdU ratio. Data are representative of the means of three individual experiments. (**f**) DNA fibre analysis from the ' + B-KO hPrimPol' DT40 cells presented as a cumulative percentage of forks at each ratio.

of RBM-A-mutant PrimPol in these cells, we expect some PrimPol would still localize to where it is required, resulting in a delay rather than a total block of fork restart.

In contrast, RBM-B mutant complemented PrimPol$^{-/-}$ cells did not display a significant increase in the mean CldU:IdU ratio when compared to cells expressing wild-type PrimPol (Fig. 5e,f). There was a slight increase in the variation of CldU:IdU ratios, however the majority of forks conformed to wild-type ratios (Fig. 5e). Again, given that PrimPol is over-expressed in these cells, a more significant effect may be observed upon mutation of the endogenous protein, with over-expression potentially masking subtle impacts on PrimPol recruitment. Nevertheless, this suggests that RBM-B is not essential for PrimPol's role in replication restart *in vivo*. Together, these results show that PrimPol's interaction with RPA, primarily mediated by RBM-A,

is important for the enzyme's role in repriming and restarting stalled replication forks following DNA damage.

**RBM-A is essential for recruitment of PrimPol to chromatin.** We previously reported that PrimPol is recruited to chromatin in response to ultraviolet damage[5]. Given the effect of mutating PrimPol's RBM-A on the enzyme's role in replication restart, we aimed to confirm if this was due to a defect in recruitment. To this end, we prepared detergent-insoluble chromatin-rich fractions from HEK-293 cells, expressing RBM-mutant PrimPol constructs, 3 h following mock or ultraviolet-C irradiation (30 J m$^{-2}$). As previously observed, wild-type PrimPol partitioned to the detergent-insoluble chromatin-enriched fraction following ultraviolet irradiation

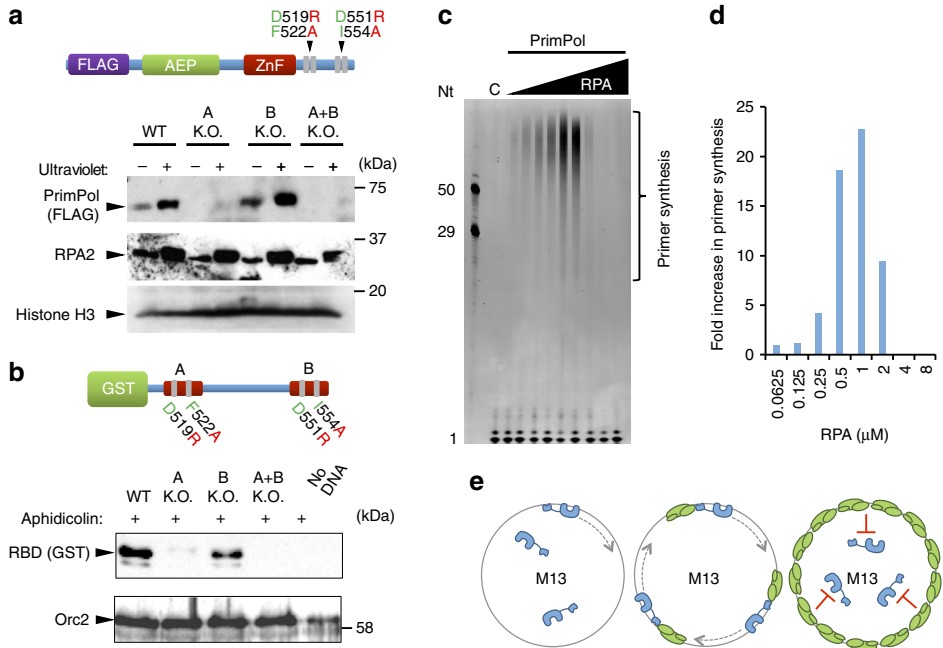

**Figure 6 | RPA recruits PrimPol to stalled replication forks *in vivo*. (a)** PrimPol's RBM-A, but not RBM-B, is critical for recruitment to chromatin. Flp-In T-Rex-293 cells transfected with WT and RBM-A and B mutant PrimPol constructs were either mock ( − ) or ultraviolet-C (30 J m$^{-2}$) ( + ) irradiated before separation into Triton X-100 (0.5%) soluble and insoluble fractions. Samples were analysed by western blot alongside whole-cell extracts. Only insoluble samples are presented here, whole-cell extracts and soluble blots can be found in Supplementary Fig. 6. **(b)** PrimPol's RBD is recruited to *Xenopus* egg extract chromatin in response to aphidicolin treatment, RBM-A is critical for this recruitment. Recombinant hPrimPol GST constructs (4 ng μl$^{-1}$) were added to *Xenopus* egg extract supplemented with sperm nuclei (3 × 10$^3$ μl$^{-1}$). Extract was treated with aphidicolin 100 μg ml$^{-1}$ and incubated at 21 °C for 80 min. Chromatin was isolated and associated proteins analysed by SDS–PAGE and western blotting using the antibodies indicated. **(c)** Low concentrations of RPA stimulate PrimPol's primase activity, high concentrations inhibit. Primer synthesis by WT hPrimPol (400 nM) on M13 ssDNA templates (20 ng μl$^{-1}$) in the presence of increasing concentrations of RPA. 'C' indicates the no enzyme control, oligonucleotide (Nt) length markers are shown on the left of the gel. **(d)** Quantification of data shown in 'c'. For each RPA concentration the fold increase in primer synthesis relative to reactions containing no RPA was calculated. Data are representative of three repeat experiments. **(e)** Schematic showing the effect of increasing RPA concentrations on PrimPol's primase activity. When no RPA is present a proportion of PrimPol binds to the M13 template and facilitates primer synthesis (left). When low/moderate concentrations of RPA are present PrimPol is recruited to the RPA/ssDNA interface causing an increase in primer synthesis activity (middle). At high RPA concentrations the M13 DNA template is fully saturated, blocking access of PrimPol to the DNA and inhibiting primer synthesis (right).

(Fig. 6a, Supplementary Fig. 6a and b). A similar increase in the level of RPA enrichment was observed in the insoluble fraction, confirming that replication forks were stalled by the damage and an increase in RPA-binding had occurred (Fig. 6a). In contrast, we found that mutation of RBM-A, either alone or in combination with RBM-B, abolished the localization of PrimPol to chromatin, both in the absence or presence of ultraviolet damage. However, mutation of RBM-B did not affect the level of enrichment of PrimPol following ultraviolet irradiation (Fig. 6a). This suggests that PrimPol's recruitment to chromatin is dependent upon its RBM-A, which is the primary mediator of the interaction of the enzyme with RPA *in vivo*.

To confirm these findings and examine the role played by the RBD of PrimPol in the recruitment of the protein to replicating chromatin, we employed a *Xenopus* synchronous cell-free extract system. We previously showed that recombinant human PrimPol accumulates on chromatin when the elongation phase of DNA replication is inhibited with aphidicolin[5]. Similarly the presence of PrimPol's RBD (480–560) is sufficient to allow recruitment of a GST fusion protein to chromatin in aphidicolin-treated extracts (Fig. 6b). RBD recruitment is severely reduced by mutation of the D519 and F522 residues within RBM-A. Mutation of the corresponding residues in RBM-B (D551, I554) also results in a modest reduction in the level of protein recruited to the chromatin, although this reduction is much less severe than that observed with the RBM-A mutations. Consistent with these

observations, a construct carrying mutations in both RBM-A and RBM-B is not detectable on the chromatin. These results demonstrate that RBM-A plays the major role in recruiting PrimPol to chromatin, with a relatively minor contribution from RBM-B.

Intriguingly, some of the key residues involved in binding of both RBM-A and RBM-B to RPA70N have been found to be mutated (F522V and I554T) in cancer patient cell lines (see COSMIC, CBioportal, CIGC repositories). We therefore generated these cancer-related PrimPol RBD mutants (F522V and I554T) in RBM-B-KO and RBM-A-KO backgrounds, respectively and analysed their binding to RPA70N using GFC (Supplementary Fig. 6c). In each case, we identify that these mutations significantly abrogate binding of the affected RBM to RPA70N, potentially suggesting that both sites play important roles in maintaining PrimPol's appropriate functions *in vivo*.

**RPA stimulates the primase activity of PrimPol.** In light of the role for RPA in recruiting PrimPol to stalled replication forks *in vivo*, we next assessed the impact of RPA on the primase activity of the enzyme *in vitro*. Using an indirect fluorescence-based primase assay, we previously identified that saturating concentrations of RPA are able to block primer synthesis by PrimPol on 60-mer poly-dT linear templates[13]. To better determine the effect of RPA on PrimPol's primase activity,

we performed direct fluorescence-based primase assays using single-stranded M13 templates in the presence of increasing concentrations of RPA. Here, we observe that sub-saturating concentrations of RPA act to significantly increase the amount of primer synthesis by PrimPol, when compared to reactions containing the enzyme only (Fig. 6c,d). Above concentrations calculated to fully coat the M13 template ($\sim$1.6 μM), the level of stimulation by RPA decreases and at higher concentrations severely inhibits primer synthesis (Fig. 6c,d). This demonstrates that lower concentrations of RPA significantly stimulate the primase activity of PrimPol, presumably by recruiting the enzyme and mediating binding to the DNA template. In contrast, high concentrations of RPA saturate the DNA template and block access of PrimPol, thus inhibiting primase activity (Fig. 6e). These results suggest that PrimPol requires a ssDNA interface adjacent to the bound RPA to be recruited to the template strand to facilitate primer synthesis.

## Discussion

Despite possessing the ability to perform TLS, recent studies suggest that PrimPol's primary role in replication restart is to reprime DNA synthesis downstream of lesions and secondary structures[9–12]. The data presented here support a model whereby PrimPol is recruited to fulfil this repriming role through its interaction with RPA (Fig. 7). This interaction is primarily mediated by residues D519 and F522 of PrimPol's RBM-A, which bind to the basic cleft of RPA70N, with RBM-B playing a supporting role in RPA-binding in vivo. In this regard, an intriguing possibility, consistent with our findings, is that RBM-B binds a second RPA molecule following initial recruitment through RBM-A in vivo, potentially contributing to the stabilization of PrimPol on the template DNA to further promote repriming. In addition to ATRIP, Mre11 and p53, we identified divergent RBM-like acidic motifs in a wide range of other DNA repair, replication and checkpoint proteins, many of which are known to interact with known to interact with RPA, for example, Werner helicase (Supplementary Fig. 7)[18].

Notably, it has been shown through crystallographic and biochemical analyses that RPA binds to ssDNA with a defined polarity[19–22]. Initial binding is mediated by the tandem DNA-binding domain A (DBD-A) and DBD-B OB folds of RPA70, forming an 8-nt binding complex. A 20–30-nt binding mode is subsequently generated by the binding of RPA's DBD-C and DBD-D[23]. This occurs in a strict 5′–3′ direction on the template strand, which likely positions the PrimPol-recruiting RPA70N domain 5′ relative to the other OB folds (Fig. 7a). This polarity suggests that PrimPol may bind downstream of RPA following recruitment through RPA70N on the leading strand. In a previous scenario[13], we speculated that PrimPol may bind upstream of RPA during TLS, due to the requirement of the ssDNA-binding ZnF domain to contact the template downstream. However, during primer synthesis the ZnF domain can access ssDNA both upstream and downstream of the AEP domain. Recent studies highlighting the importance of PrimPol's primase activity in vivo[9–12], coupled with the recruitment of the enzyme via RPA70N shown here, argue that PrimPol more likely binds downstream of RPA, with the ZnF bound to ssDNA upstream of the AEP domain, during primer synthesis (Fig. 7b).

PrimPol displays low processivity, only extending primers 1–5 nt in a single-binding event[10]. This processivity is in part regulated by the ZnF domain, which serves as a 'counting mechanism' to limit primer extension by the AEP domain[10], as has been observed with other primases[24]. The ZnF and AEP domains therefore likely form a hinge-like structure with the ZnF domain limiting extension by PrimPol following initial primer

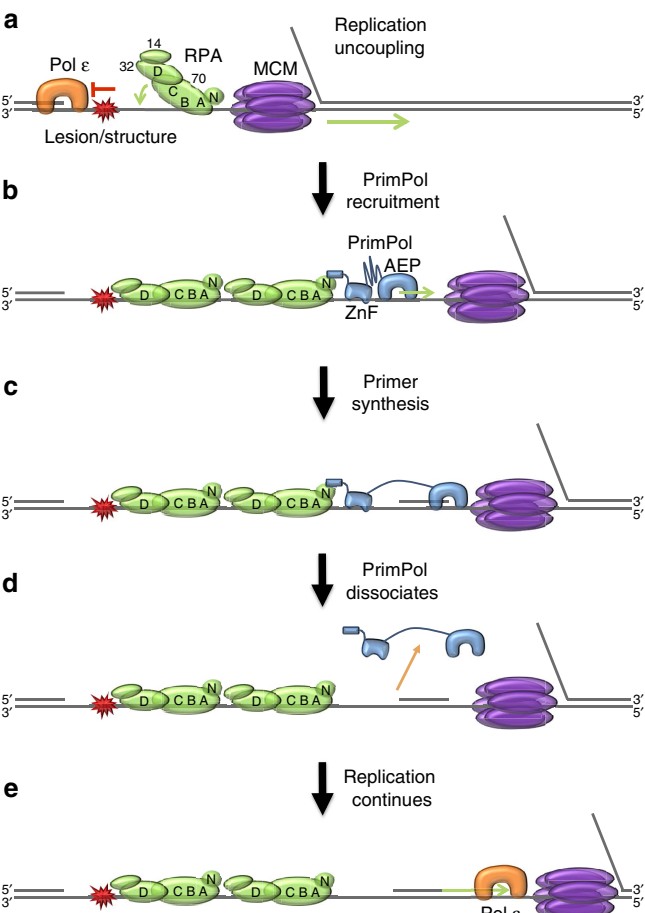

**Figure 7 | Model for PrimPol recruitment to stalled replication forks by RPA.** (**a**) Unrepaired DNA damage lesions, or DNA secondary structures, in the leading strand template lead to stalling of polymerase ε. This causes uncoupling of replication, generating ssDNA downstream of the DNA damage lesion/structure and facilitating binding of RPA. Note that for simplicity other replisome components and lagging strand synthesis machinery are not shown. (**b**) PrimPol is recruited to the ssDNA interface uncovered by the replicative helicase through the interaction of its RBM-A with RPA70N. This interaction is stabilized by the binding of the ZnF and AEP domains to ssDNA. (**c**) PrimPol catalyses the synthesis of a new DNA primer, before further extension is prevented by the restraining effect of the RPA interaction and ZnF domain, coupled with the enzyme's low processivity. (**d**) Unable to continue with primer extension, PrimPol dissociates from the template strand. Re-binding upstream is prevented by RPA. (**e**) The nascent primer is utilized by the replicative polymerase for continued DNA replication. This leaves behind a short RPA-coated ssDNA region opposite the lesion to be filled in by template switching or TLS.

synthesis[10,24]. Given that PrimPol is recruited by RPA in vivo, it is likely that the enzyme initially binds the ssDNA downstream of RPA in a 'closed hinge' mode (Fig. 7b). Primer synthesis and polymerization then proceed until PrimPol reaches its maximum open conformation, dictated by the ZnF domain and interaction with RPA, which thereby prohibit further extension (Fig. 7c,d). The newly synthesized primer can then be utilized by the replicative polymerase for continued extension (Fig. 7e).

We previously reported that, in contrast to the effect on replicative polymerases, RPA acts to inhibit the polymerase activity of PrimPol[13]. This phenomenon may be explained by the polarity of RPA when bound to ssDNA, in addition to the protein's interaction with PrimPol. It has been suggested that replicative polymerases are able to readily displace RPA from

DNA because they encounter the protein from the 3′ side[25]. As the replicative polymerase synthesizes DNA, moving 3′-5′ on the template strand, it would first encounter the relatively weakly bound RPA32 DBD-D and RPA70 DBD-C, before RPA70 DBD-B and A. By approaching RPA in this orientation and making specific protein–protein interactions, the replicative polymerase may shift the equilibrium from the stronger 20–30-nt RPA-binding mode to the weaker 8-nt mode[23], and in turn, the more weakly bound RPA can be displaced by further DNA synthesis. In contrast, recruitment of PrimPol to the 5′ side of RPA would result in the enzyme moving away from the protein, making it unable to displace RPA in the same manner as canonical polymerases.

In addition, we show that RPA stimulates the primase activity of PrimPol at sub-saturating concentrations. However, when the template is fully coated with RPA, the primase activity of PrimPol is inhibited. This suggests that PrimPol requires a ssDNA interface adjacent to RPA to be efficiently recruited for priming. Given that PrimPol likely binds downstream of RPA on the leading strand during replication, this ssDNA interface could be formed following uncoupling of leading and lagging strand replication upon stalling at a DNA lesion or secondary structure[26]. Continued unwinding of duplex DNA by mini-chromosome maintenance protein (MCM) may generate the leading strand ssDNA interface necessary for PrimPol to reprime, following recruitment by RPA. It was recently reported that the mitochondrial replicative helicase Twinkle is able to stimulate DNA synthesis by PrimPol[27], potentially suggesting that replicative helicases can facilitate synthesis by the enzyme *in vivo*. However, the exact interplay between RPA, PrimPol and other PrimPol-interacting partners, such as PolDIP2 requires further examination[28]. The necessity of a ssDNA interface for PrimPol activity, in conjunction with the enzyme's inability to displace RPA, may act as an important regulatory mechanism to prevent unscheduled primer synthesis during replication. Intriguingly, it has been hypothesized that recruitment of DNA damage response proteins to RPA70N may be regulated by phosphorylation of RPA32C (ref. 18). In support of this, it has been shown that binding of Mre11 and Rad9 to RPA is increased upon RPA32C phosphorylation[29,30], although it remains to be determined if this is also the case for PrimPol.

Together, the findings presented here describe the molecular basis of PrimPol's interaction with RPA and provides insights into its biological roles. We found that the PrimPol-RPA interaction, mediated primarily by RBM-A, is essential for PrimPol recruitment and its function as a repriming enzyme during DNA replication. Notably, mutations of critical residues in both RBMs have been identified in the genomes of some cancer patient cell lines and we have shown that these mutations are sufficient to abrogate the functionality of their respective RBMs. Further studies are underway to more precisely define how PrimPol is recruited to stalled replication forks and regulated by interactions with other fork proteins to better understand the critical roles played by PrimPol in the restart of stalled replication forks.

## Methods

**Expression of human PrimPol and RPA truncation variants.** Full-length PrimPol was expressed and purified as previously described[5]. Briefly, PrimPol amino acids 480–560 (PrimPol$_{480-560}$) was cloned into pET28a by polymerase chain reaction using wild-type PrimPol as a template via standard methods (primers 1 and 2, Supplementary Table 1). PrimPol$_{480-560}$ was expressed in BL21(pLysS) cells overnight at 25 °C and purified using Ni Sepharose (Qiagen), followed by Q Sepharose (GE Healthcare) and gel filtration using a Superdex 75 10/300 GL column (GE Healthcare) according to the manufacturer's instructions. PrimPol residues 480–546 (PrimPol$_{480-546}$), PrimPol D514R, D518R, D519R (PrimPol$_{RBM-A-KO}$), PrimPol D514R, D518R, D519R, D551R, I554A, I555A (PrimPol$_{RBM-A-KO/RBM-B-KO}$), PrimPol F522V on an RBM-B-KO background and PrimPol I554T on an RBM-A-KO background were cloned by site-directed

mutagenesis (primers 3–12, Supplementary Table 1). PrimPol 480–546 with the D514R, D518R, D519R mutations (PrimPol$_{480-546/RBM-A-KO}$) was also cloned, using the 480–546 construct DNA as a template. All these proteins were expressed and purified as described for PrimPol$_{480-560}$.

The RPA trimeric complex was expressed and purified as previously described[13]. Briefly, RPA was expressed in BL21 (DE3) *E. coli* cells harbouring p11d-tRPA for 12 h at 15 °C. Following harvesting, cells were lysed in buffer containing 500 mM NaCl, 100 mM spermidine, 4 mg ml$^{-1}$ lysozyme and 1 mM phenylmethylsulfonyl fluoride and clarified by centrifugation. Protein purification followed a 5-step procedure using Affi-Gel Blue (Bio-Rad), HiTrap heparin HP, HiTrap SP FF and MonoQ HR 5/5 columns (GE Healthcare), before size-exclusion chromatography (GE Healthcare). Following purification, RPA was snap-frozen in liquid nitrogen and stored at −80 °C in buffer containing 30 mM Tris–HCl (pH 7.5), 300 mM NaCl, 2 mM TCEP and 10% (v/v) glycerol.

RPA70N (RPA70$_{1-120}$) was cloned as described previously[14,31]. Briefly, wild-type and mutant RPA70N pET15b constructs were transformed into BL21 (DE3) *E. coli* cells. Cultures were grown at 37 °C in Terrific Broth (RPI) supplemented with 4 g l$^{-1}$ glycerol and maintained at pH 7.2 using a BioFlo 3,000 bioreactor (New Brunswick Scientific). Isotopically enriched protein samples for NMR were grown and expressed in M9 medium containing 0.5 g l$^{-1}$ $^{15}$N-NH$_4$Cl (ref. 15). Protein expression was induced overnight at 18 °C by addition of 1 mM IPTG. Cells were lysed by sonication in a buffer containing 20 mM Tris-Cl (pH 7.5), 300 mM NaCl and 10 mM imidazole, then loaded onto a HisTrap HP column (GE) using a Bio-Rad NGC FPLC. A gradient of buffer with 300 mM imidazole was used to elute the protein. After dialysis to remove the imidazole, the polyhistidine tag was cleaved with thrombin for 1 h and the sample repassed over the Histrap column. It was then polished using size-exclusion chromatography with a Superdex 75 column in a buffer containing 20 mM HEPES (pH 7.2), 80 mM NaCl, 5 mM dithiothreitol (DTT). Final protein concentration was measured using a nanophotometer.

The RPA70N$^{E7R}$ variant that readily forms crystals with basic-site ligands was utilized in the experiments shown here[15]; the properties of this protein variant are not affected in any way apart from in its crystal lattice contacts. Protein concentrations were determined based on absorbance at 280 nm corrected with the protein-specific extinction coefficient. Extinction coefficient values for each of the recombinant proteins were calculated using ProtParam tool (ExPASy).

RBM-A (510–528) and RBM-B (546–560) peptides for NMR were synthesized (GenScript), purified with a Waters Delta 600 HPLC using a Proto 300 C4 column (Higgins Analytical, Inc.) and confirmed using mass spectrometry. The PrimPol$_{514-528}$ peptide used for co-crystallization experiments was synthesized (Genscript) and used as supplied

**Nuclear magnetic resonance (NMR) methods.** 15N-1H HSQC experiments were performed at 25 °C on a Bruker Avance III 800 or 900 MHz NMR spectrometer with a cryogenically cooled probe. Spectra were acquired for 100 μM samples of 15N-enriched RPA70N or 15N-enriched PrimPol$_{480-560}$ alone and in the presence of 200 μM unlabelled binding partner. All samples were equilibrated in a buffer containing 20 mM HEPES (pH 7.5), 80 mM NaCl, 2 mM DTT and 5% deuterium oxide.

**Analytical size-exclusion chromatography.** Protein interactions were analysed by size-exclusion chromatography on a Superdex S75 10/300 GL gel filtration column (GE Healthcare). The column was calibrated using albumin (67,000 Da), ovalbumin (43,000 Da), chymotrypsinogen A (25,000 Da), ribonuclease A (13,700 Da) and aprotinin (6,512 Da). The protein was loaded at 0.5 ml min$^{-1}$. Retention volume of the proteins were plotted against the molecular weight of each protein to reliably predict protein molecular weights. The column was pre-equilibrated in a buffer containing 50 mM Tris–HCl (pH 7.5), 100 mM NaCl and 2 mM TCEP that had been sterile-filtered using a 0.2 μm pore size vacuum filtration system (Nalgene). In total, 0.5 ml of protein was loaded at a concentration of 35 μM. Protein interactions were determined by a shift in the chromatograph peaks relative to individual protein peaks.

**SEC multiangle light scattering.** SEC-MALS was performed on an AKTA Purifier FPLC system (GE) connected to an Agilent Technologies 1200 Series refractive index Detector and a Wyatt Technologies Dawn Helios 8+ MALS unit. A Superdex 75 increase 10/300 GL (24 ml) column was equilibrated in running buffer consisting of 20 mM HEPES (pH 7.1), 80 mM NaCl, 0.5 mM TCEP. The flow was maintained at a consistent 0.5 ml min$^{-1}$ and sample injections of 100 μl from a static loop were initiated at the 0 ml point of each run. Ultraviolet, refractive index, light scattering (LS) and Quasi-Elastic LS values were recorded using ASTRA 6.1 (Wyatt) software. Data were collected using samples of RBD at 185 μM with RPA70N E7R added at 0, 1, 2 and 4× molar ratios. Estimated molecular weights for RBD and its saturated complexes were calculated using the Zimm algorithm surrounding the peak maximum.

**Crystallization and X-ray structure determination.** Crystals of the RPA70N-PrimPol complex were grown at 293 K by vapour diffusion as sitting drops. The protein complex was screened at a 2.5:1 ratio of 1.75 mM PrimPol$_{514-528}$

peptide: 0.70 mM of RPA70N$^{E7R}$ in drops containing 0.5 µl of protein complex mixed with 0.5 µl of crystallization buffer (0.2 M ammonium acetate 0.1 M sodium acetate 4.5 20% w/v PEG 3350). Before data collection, crystals were cryoprotected by soaking in mother liquor containing 20% ethylene glycol. X-ray diffraction data of 1.542 Å was collected in-house at 100 K using a Rigaku MicroMax 007-HF. The diffraction data were processed with SCALA[32] with additional processing by programs from the CCP4 suite[33].

Crystals of the RPA70N-PrimPol complex were grown at 293 K by vapour diffusion as sitting drops. The protein complex was screened at a 1:1 ratio with 700 µM of each of RPA70N$^{E7R}$ and PrimPol$_{480-560}$; 0.5 µl of protein complex was mixed with 0.5 µl of crystallization buffer (0.2 M imidazole malate (pH 6.0), 30% (w/v) PEG 4000). Before data collection, crystals were soaked in mother liquor containing 20% ethylene glycol. X-ray diffraction data of 0.914 Å were collected at 100 K using a synchrotron source at station I03 Diamond Light Source, Didcot, UK. The diffraction data were processed with xia2 (ref. 34) with additional processing by programs from the CCP4 suite[33]. The statistics for data processing are summarized in Table 1. For both models, initial phases were obtained by molecular replacement with PHASER[35] (using RPA70N$^{E7R}$ (4IPC)[15] as a search model). Iterative cycles of model building and refinement were performed using Coot[36] and Phenix[37]. A final refined model at 2.0 Å resolution, with an $R_{factor}$ of 18.73% and $R_{free}$ of 22.86% was obtained for the RPA70N-PrimPol$_{514-528}$ peptide complex. The Ramachandran statistics for this complex place 97.9% of residues in the favoured region and 2.1% in the allowed region. For the RPA70N-PrimPol complex a refined model at 1.28 Å resolution, with an $R_{factor}$ of 15.37% and $R_{free}$ of 17.85% was obtained with Ramachandran statistics of 98.6% of residues in the favoured region and 1.4% in the allowed region. Structural images were prepared with CCP4mg (ref. 38). Stereo images for portions of the electron density of RPA70N-PrimPol514–528 and RPA70N-PrimPol480–560 are shown in Supplementary Fig. 8. The structures of the RPA70N-PrimPol$_{514-528}$ peptide complex and the RPA70N-PrimPol$_{480-560}$ complex are deposited in the Protein Data Bank under accession codes 5N85 and 5N8A, respectively.

**Circular dichroism.** PrimPol RBD samples for CD were equilibrated in 20 mM HEPES (pH 7.5), 80 mM NaCl and 2 mM DTT, and then diluted 1:10 with ultrapure water to a final concentration of 20 µM. A JASCO J-810 spectro-photometer equilibrated at 25 °C was used to collect five scans over the spectral width 190–250 nm. Molar ellipticity was calculated based on the final protein concentration of 20 µM.

**Dynamic light scattering.** Light scattering experiments were performed using a Wyatt Technology DynaPro NanoStar instrument. PrimPol RBD was equilibrated in 20 mM HEPES (pH 7.5), 80 mM NaCl and 2 mM DTT at a concentration of 200 µM. A 5 µl sample was then equilibrated in a COC cuvette at 25 °C for 5 min before acquisition. Ten data points were acquired and fitted using the coils protein shape model using Wyatt Dynamics software. The resulting regularization graph was plotted as a function of %mass and Mw-R calculated based on observed sample radius.

**Isothermal titration calorimetry.** Isothermograms were recorded using a MicroCal VP-ITC instrument. In total, 10 µl injections of 400 µM PrimPol RBD (either WT or variants) were added to a 1.4 ml cell with RPA70N at 20 µM. The system was equilibrated for 5 min between injections. Both proteins were dialyzed in the same pool of 20 mM HEPES (pH 7.5), 80 mM NaCl and 3 mM β-mercaptoethanol buffer. Dissociation constants were calculated with MicroCal Origin software using a single-site binding model.

**Fluorescence-based M13 primase assay.** Full-length PrimPol (400 nM) was incubated in 20 µl reactions containing 10 mM Bis–Tris-Propane-HCl (pH 7.0), 10 mM MgCl$_2$, 1 mM DTT, 250 µM dNTPs, 2.5 µM FAM dNTPs (dATP, dCTP, dUTP) and 20 ng µl$^{-1}$ single-stranded M13 template, at 37 °C for 15 min. Individual reactions were supplemented with increasing concentrations of RPA (0, 0.0625, 0.125, 0.25, 0.5, 1, 2, 4 and 8 µM) before the addition of PrimPol. Following primer synthesis, remaining free FAM dNTPs were removed using an Oligo Clean and Concentrator kit (Zymo Research) according to the manufacturer's instructions. Eluted primers were supplemented with loading buffer (95% formamide with 0.25% bromophenol blue and xylene cyanol dyes; total volume 20 µl). Samples were boiled and resolved on a 15% polyacrylamide/7 M urea gel for 90 min. Products were visualized on an FLA-5100 imager.

**Maintenance and generation of HEK-293 Flp-In T-Rex cells.** HEK-293 Flp-In T-REx (Invitrogen) cells were cultured in DMEM containing 10% fetal calf serum (FCS), 1% L-glutamine and 1% PenStrep. For the generation of stable inducible N-terminal FLAG-tagged PrimPol HEK-293 Flp-In T-REx, cells were grown in medium containing 15 µg ml$^{-1}$ Blasticidin (Invitrogen) and 100 µg ml$^{-1}$ Zeocin before transfection. Cells were transfected with pOG44 plasmid and pcDNA5/FRT/TO plasmid (1:9 ratio) encoding FLAG-PrimPol (WT, D519R/F522A, D551R/I554A, D519R/F522A/D551R/I554A and PrimPol$^{480-560}$) using Lipofectamine 2000 following the manufacturer's instructions. pcDNA5/FRT/TO constructs

encoding N-terminal FLAG-PrimPol and FLAG-PrimPol$^{480-560}$ were generated by standard PCR and cloning procedures (primers 13, 14 and 2, Supplementary Table 1). RBM-mutant N-FLAG-PrimPol constructs were produced by site-directed mutagenesis (primers 15–18, Supplementary Table 1). After 48 h of transfection, selective medium containing 15 µg ml$^{-1}$ Blasticidin and 100 µg ml$^{-1}$ Hygromycin (Invitrogen) was added. Selective medium was replaced every 2–3 days, until resistant clones appeared. Clones were then pooled, expanded and stocks made.

**Co-immunoprecipitation in FLAG-PrimPol HEK-293 cells.** HEK-293 Flp-In T-REx cells engineered for inducible expression of FLAG-PrimPol (WT, D519R/F522A, D551R/I554A, D519R/F522A/D551R/I554A and PrimPol$^{480-560}$) were grown in the presence or absence of doxycycline (10 ng ml$^{-1}$) 24 h before collecting. Cell pellets were resuspended in 1 ml lysis buffer (150 mM NaCl, 30 mM Tris–HCl (pH 7.5), 0.5% NP40, 2.5 mM MgCl$_2$, 100 µg ml$^{-1}$ DNase I) and incubated at 4 °C for 30 min. The resulting lysate was clarified by centrifugation at 10,000 g for 10 min at 4 °C. The supernatant was retained (sample taken as 'input'), added to 100 µl pre-washed anti-FLAG magnetic beads (Sigma) and incubated at 4 °C overnight. Unbound material was removed and the beads were washed 3 × 5 min with 1 ml wash buffer (Lysis buffer without DNase I and 0.1% NP40). Three successive 5 min elutions were performed using 200 µl elution buffer (25 mM Tris–HCl (pH 7.5), 150 mM NaCl, 1 mM phenylmethyl sulphonyl fluoride and 200 µg ml$^{-1}$ 3 × FLAG peptide (Sigma). Eluted samples were boiled in Laemmli buffer and analysed by western blot using the following antibodies: Anti-FLAG (Sigma F3165; 1:1,000 dilution), Anti-RPA2 (Calbiochem NA18; 1:500 dilution), horseradish peroxidase (HRP) conjugated Anti-mouse IgG (Abcam ab6728; 1:5,000 dilution). Uncropped versions of all Western blots can be found in Supplementary Fig. 9.

**Triton X-100 fractionation of HEK-293 cells.** HEK-293 cellular fractionation was performed as previously described[5]. Briefly, protein expression was induced (10 ng ml$^{-1}$ doxycycline, 24 h) in HEK-293 Flp-In T-REx cells stably transfected with various FLAG-PrimPol constructs (WT, D519R/F522A, D551R/I554A and D519R/F522A/D551R/I554A). The following day cells were either mock or ultraviolet-C (30 J m$^{-2}$) irradiated and allowed to recover for 3 h. Cells were harvested and pellets resuspended in cytoskeletal buffer (100 mM NaCl, 300 mM sucrose, 3 mM MgCl$_2$, 10 mM PIPES (pH 6.8), 1 mM EGTA, 0.2% Triton X-100 and protease and phosphatase inhibitors (Roche), followed by incubation on ice for 5 min. Samples were then centrifuged at 16,000 g for 10 min. Supernatant was retained as the soluble fraction. The insoluble pellet was washed three times in PBS and boiled in Laemmli buffer. Whole-cell extract, soluble and insoluble, samples were analysed by western blot using Anti-FLAG, Anti-RPA2 and Anti-mouse IgG antibodies sourced and used as described above, in addition to Anti-Histone H3 (Abcam ab1791; 1:5,000 dilution) and HRP-conjugated Anti-Rabbit IgG (Abcam ab6721; 1:3,000 dilution).

**Chromatin isolation from Xenopus egg extract.** Demembranated sperm nuclei were prepared by lysolecithin treatment as previously described[39]. Briefly, Xenopus sperm nuclei in 1 ml of SuNaP (250 mM sucrose, 75 mM NaCl, 0.15 mM spermine, 0.5 mM spermidine) were treated with 50 µl of 10 mg ml$^{-1}$ lysolethicin (per 10$^7$ nuclei) for 5 min at room temperature. Reaction was stopped with 5 ml SuNaP + 3% BSA. Nuclei were washed twice in SuNaP and resuspended at a concentration of 1 × 10$^5$ nuclei per µl in SuNaP/glycerol (final glycerol concentration 30%) and snap-frozen and stored in liquid nitrogen. Preparation of Xenopus egg extracts and the isolation of chromatin from egg extract were carried out as previously described[40]. Briefly, unfertilized eggs were dejellied, washed and activated with ionophore A23187. After washing in extraction buffer (XB: 10 mM 4-(2-hydroxyethyl)piperazine-1-ethanesulfonic acid (HEPES-KOH), pH 7.7, 100 mM KCl, 0.1 mM CaCl$_2$, 1 mM MgCl$_2$, 50 mM sucrose) eggs were packed by brief centrifugation at 5,000 g and then, after removal of excess buffer, crushed by centrifugation at 15,000 g for 10 min (4 °C). The cytoplasmic layer was supplemented with aprotinin (10 µg ml$^{-1}$), cytochalasin B (50 µg ml$^{-1}$), creatine phosphate (30 mM) and creatine phosphokinase (150 µg ml$^{-1}$) and centrifuged for 10 min at 60,000 g (4 °C) in a Beckman Optima TLA-55, to generate the replication-competent supernatant fraction.

For chromatin isolation, 50 µl of extract containing sperm chromatin (5,000 nuclei per µl) was diluted with XB buffer containing 0.25% Triton X-100 and centrifuged through 750 mM sucrose (in XB) at 5,000 g (10 min, 4 °C). The cytoplasmic/sucrose interface was washed twice with XB/Triton X-100, the supernatant removed and the chromatin pellet washed with XB/Triton X-100. After centrifuging at 10,000 g (5 min, 4 °C), the wash buffer was removed and the chromatin pellet resuspended in SDS–PAGE buffer.

Western blot analysis was performed using the following antibodies; Anti-GST (Abcam ab92; 1:2,000 dilution), Anti-Orc2 (gift from Julian Blow; 1:2,000 dilution), HRP-conjugated Anti-mouse IgG (DAKO P0260; 1:5,000 dilution) and HRP-conjugated Anti-rabbit IgG (DAKO P0448; 1:5,000 dilution).

**DNA fibre assays in PrimPol$^{-/-}$ DT40 cells.** DT40 cells were cultured in RPMI 1640 medium containing 10% FCS, 1% chicken serum, 10 µM

β-mercaptoethanol, 1% L-glutamine and 1% PenStrep. Mutant DT40 cell lines were derived from DT40 Clone 653 from Prof. S. Takeda's group (Kyoto University). PrimPol$^{-/-}$ DT40 cells (previously generated[5]) were stably complemented with pCI-neo plasmid encoding WT PrimPol, PrimPol$^{D519R/F522A}$ and PrimPol$^{D551R/I554A}$ by electroporation as previously detailed[5]. pCI-neo constructs-encoding RBM-mutant PrimPol were generated by site-directed mutagenesis (primers 15–18, Supplementary Table 1). Positive clones were selected using medium containing 2 mg ml$^{-1}$ G418 (Sigma) and expression was confirmed by western blot using Anti-PrimPol (raised against recombinant purified PrimPol, 1:1,000), Anti-α-Tubulin (Sigma T5168, 1:3,000 dilution), and HRP-conjugated Anti-Rabbit IgG and Anti-Mouse IgG (sourced and used as described above). All DNA fibre analysis was performed as described previously[5] in triplicate. Briefly, DT40 cells were pulse labelled with 25 μM CldU for 20 min before ultraviolet irradiation with 20 J m$^{-2}$ and labelling with 250 μM IdU for a further 20 min. Subsequently, cells were resuspended in PBS and diluted 1:1 with unlabelled cells. The mixture was spotted onto glass Superfrost slides, lysed with buffer containing 0.5% SDS, 200 mM Tris–HCl (pH 5.5) and 50 mM EDTA, and tilted to allow spreading of DNA. Slides were fixed in 3:1 methanol/acetic acid and stored at 4 °C. Immunolabelling was performed after rehydration, denaturation of DNA and fixing in 4% paraformaldehyde, using anti-rat BrdU (Abcam ab6326, 1:1,000 dilution), anti-mouse BrdU (Bection Dickinson 347580, 1:500 dilution), and secondary Alexa Fluor 488-labelled anti-rat (Abcam ab150157, 1:250 dilution) and Alexa Fluor 594-labelled anti-mouse (Abcam ab150116, 1:250 dilution).

**Yeast two-hybrid assay.** Full-length PrimPol and its C-terminal RBM domain (PP-RBM—a.a. 480–560) were cloned into NdeI site of the pGADT7 vector using polymerase chain reaction with wild-type PrimPol as a template, and T4 polymerase to process the DNA ends. PrimPol mutants were prepared by site-directed mutagenesis. RPA70N (a.a. 1–120) was cloned into NdeI site of the pGBKT7 vector. Plasmids containing the GAL4 activation domain (pGADT7) fused to the PrimPol variants or the empty vector were transformed into the *Saccharomyces cerevisiae* strain PJ69-4a. Plasmid containing the GAL4 DNA-binding domain (pGBKT7) fused to RPA70N or the empty vector were transformed into PJ69-4α strain. The haploid strains were mated on a YPD plate and replica plated on selective medium lacking leucine and tryptophan. The resulting diploid strains were grown to A600 ∼1 and spotted as 10-fold serial dilutions on media lacking leucine, tryptophan, histidine or adenine. A total of 1 mM 3-Amino-1,2,4-triazole (3AT) was added to decrease the background HIS3 expression. Plates were scanned after 3 days of incubation at 30 °C

**Data availability.** All data are provided in full in the results section and the Supplementary Information accompanying this paper. Atomic coordinates and structure factors have been deposited in the Protein Data Bank with accession codes 5N85 and 5N8A for the RPA70N-RBD-A and B complexes, respectively.

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

## Acknowledgements

We thank D. Reid Putney and Markus Voehler for assistance with NMR data acquisition and presentation. We also thank Dr M. Roe for assistance with X-ray data collection. A.J.D.'s laboratory supported by grants from the Biotechnology and Biological Sciences Research Council (BBSRC: BB/H019723/1, BB/M008800/1 and BB/M004236/1). T.A.G. and B.A.K. were supported by University of Sussex and BBSRC (1098524) PhD studentships respectively. Work in the Chazin lab was supported by U.S. National Institutes of Health grants T32 CA009582 and F32 GM116302 to A.E. and R01 GM65484 and R35 GM118089 to W.J.C. NMR instrumentation was supported from the NSF (0922862), NIH (S10 RR025677) and Vanderbilt University matching funds. Funding for open access charge: Research Councils UK (RCUK).

## Author contributions

A.J.D. designed the project and directed the experimental work. T.A.G., N.C.B., A.E., H.D.L. and W.J.C. also participated in project design. T.A.G. purified full-length PrimPol, CTD cancer mutants and RPA, generated the HEK-293 and DT40 stable cell lines, performed the *in vitro* primase assay, immunoprecipitation experiments, DNA fibre analysis and triton fractionation, and developed the model. A.E. and N.C.B. identified and validated RBM interactions. N.C.B., A.E. and B.A.K. designed and purified samples of RBM-A/B peptides, RPA70N E7R and RBD and its mutants. N.C.B. performed X-ray data collection processing and model building. T.A.G., N.C.B., B.A.K. and A.E. performed analytical size-exclusion chromatography. A.E. and W.J.C. designed and analysed, and A.E. performed, NMR, ITC, CD, dynamic light scattering and SEC-MALS experiments. P.K. performed the yeast two-hybrid experiments. E.M.T. and H.D.L. performed the chromatin isolation from *Xenopus* egg extracts and subsequent interaction studies using this system. L.J.B. generated the wild-type human PrimPol stable DT40 cell line. T.A.G. and A.J.D. wrote the manuscript. N.C.B., A.E., B.A.K. and W.J.C. assisted in writing the manuscript.

## Additional information

**Competing interests:** The authors declare no competing financial interests.

