## [Peer review file · Nature Communications]

Reviewers' comments:

Reviewer #1 (Remarks to the Author):

In this work, Guillian et al describe the biochemical and structural characterization of the interaction between the human primase-polymerase PrimPol and the single stranded DNA binding protein RPA. Their group and others have previously shown that these interaction is essential to recruit PrimPol to sites of DNA damage and initiate primer synthesis required for restart of the replication fork at sites of DNA damage. In this work, they reveal the molecular details of this interaction by a nice set of complementary experiments. They identify two RPA binding motifs in the C-terminal region of PrimPol, but then also show that only one of these is required for the in vivo function of PrimPol in replication restart. They furthermore provide an attractive model for how this interaction may function during re-priming/replication restart. The manuscript is well written and the data is presented in a clear manner. Overall, this work provides a significant progress for the understanding of PrimPol's role in maintenance of genome integrity.

I only have a few comments/questions

Main comments

1) The authors show that the two binding motifs RBM-A and RBM-B both bind RPA70N with similar affinities (SupFig 4B-C) and through similar interactions (fig 1D & 2E) in vitro. Yet they show that in vivo only RBM-A is required. Could they speculate why this may be the case, i.e. is it sequence dependent or structure dependent though interactions/clashes with other domains?

(A potentially interesting future experiment would be to swap the two motifs and monitor the effect in vivo.)

2) The GFC analysis of the WT RBD domain shows an unusual early elution for such a small protein. They present dynamic light scattering and circular dichroism data to indicate that the protein is monomeric in solution. Yet, it would be more convincing if they could perform a SEC-MALS (Size Exclusion Chromatography with Multi-Angle Light Scattering) experiment that will provide an accurate measure for the molecular weight.

3) When comparing the GFC from wild type RBD-RPA70N complex with the mutant versions RBD-Ak-o and RBD-Bk-o, it appears that the wild type RBD-RPA70N complex elutes at ~10.2ml, while the mutant version elute closer to 11ml. Is it possible that two RPA70N molecules can bind to one wild type RBD? A SEC-MALS experiment could easily resolve this (see also above).

4) In Fig 4 & 5, they show that the RBD-A motif is required for interaction with RPA and replication restart after DNA damage. Are the RBD-Ak-o (D519R/F522A) cells also more sensitive to UV light and/or DNA damaging agents?

Minor comments

- 5) Page 5, line4: "...interacts directly with RPA70N". Please indicate the boundaries of the construct (i.e. residues 1-120) .
- 6) Page 6 third section: RPA70NE7R. Please state in the text if the E7R mutation is close to, or distal from the RBD binding site, and if the mutation could affect the binding to RBD-A.
- 7) Page 7, second section: "PrimPol-RBD" and "PrimPol480-560" are both used to indicate the same construct. Please use only one (and indicate residues numbers if using PrimPolRBD).
- 8) Page 8, first section, line 4. Change "70N" to "RPA70N"
- 9) Page 8, second section, last 2 lines. "These interactions are of paramount importance in the binding of PrimPol to RPA70N". This is an odd statement given that later on in the manuscript it is shown that the RBD-B motif is dispensable for PrimPol-RPA interaction in vivo. Please change or remove.

Reviewer #2 (Remarks to the Author):

The paper by Guillian and colleagues builds on previous work by the same group and others (Guillian et al, NAR, 2014; Wan et al, EMBORep, 2013), which showed that PrimPol recruitment to replication forks depends on the ssDNA-binding protein RPA, mediated by PrimPol's C-terminal domain.

The novelty of this work lies in the identification of two RPA-binding motifs in PrimPol's C-terminal domain, the biophysical characterisation of the interaction and cell-based evidence of the physiological relevance of the interaction.

The experiments appear to have been correctly executed and the experimental evidence presented is of good quality. The work addresses a narrow experimental point, and as such it will be of direct interest predominantly to PrimPol biologists.

Specific points:

If PrimPolRBD contains two RPA-binding domains of equal affinity, a PrimPol-RPA complex of 1:2 stoichiometry should be observable, in the GFC experiments or by other means.

"Similar to PrimPolRBD, PrimPolA-K.O. eluted as a single defined multimeric complex from GFC (Fig. 3a)."

What is the multimeric complex referred to here? The data in SuppIFig 1 show that PrimPolRBD is monomeric in solution.

"Continuous electron density covers the entirety of RPA70NE7R and of the 15 residues in the synthesised PrimPol514-528 peptide, 9 residues (PrimPol517-525) are visible in the electron density maps. Within this short peptide, residues aspartate 516 to glutamate 524 are α -helical in content."

Only residues 517-525 are visible in the map, but aspartate 516 is helical? Please clarify.

The authors should report the PDB IDs for deposition of crystallographic models and structure factors.

Reviewer #1 (Remarks to the Author):

In this work, Guilliam et al describe the biochemical and structural characterization of the interaction between the human primase-polymerase PrimPol and the single stranded DNA binding protein RPA. Their group and others have previously shown that these interaction is essential to recruit PrimPol to sites of DNA damage and initiate primer synthesis required for restart of the replication fork at sites of DNA damage. In this work, they reveal the molecular details of this interaction by a nice set of complementary experiments. They identify two RPA binding motifs in the C-terminal region of PrimPol, but then also show that only one of these is required for the in vivo function of PrimPol in replication restart. They furthermore provide an attractive model for how this interaction may function during re-priming/replication restart. The manuscript is well written and the data is presented in a clear manner. Overall, this work provides a significant progress for the understanding of PrimPol's role in maintenance of genome integrity.

Main comments

1) The authors show that the two binding motifs RBM-A and RBM-B both bind RPA70N with similar affinities (SupFig 4B-C) and through similar interactions (fig 1D & 2E) in vitro. Yet they show that only RBM-A is required in vivo. Could they speculate why this may be the case, i.e. is it sequence dependent or structure dependent though interactions/clashes with other domains? (A potentially interesting future experiment would be to swap the two motifs and monitor the effect in vivo.)

In the revised manuscript, we have included SEC-MALS data (described below in more detail) which are consistent with PrimPol's RBD binding to two

RPA70N molecules at the same time in vitro. We have also included yeast-two hybrid experiments revealing that in the context of the full-length enzyme, both RBM-A and RBM-B must be mutated to completely abolish the interaction with RPA70N. This suggests that the full-length enzyme is capable of binding to RPA via both RBM-A and B.

Nevertheless, it is clear from the in vivo data, that in the proper context of the vertebrate cell, RBM-A is the primary binding site for PrimPol's interaction with RPA. This data includes, a complete loss of binding to RPA, slowed replication fork speeds after UV damage, and loss of recruitment to chromatin for both the full-length enzyme and RBD, upon RBM-A mutation. Despite the effects of RBM-B mutation being much more subtle, there was a reduction in RPA binding, compared to the wild-type protein, in the absence of a functional RBM-B, but unmodified RBM-A, in vivo. Additionally, any subtle effect on replication fork speed following mutation of RBM-B, which could arise upon mutation of the endogenous protein, may be masked here due to the high levels of PrimPol overexpression in these cells (this is now clarified in the main text). These data suggest that binding of RBM-B to RPA is dependent on a functional RBM-A, however, RBM-A is able to bind RPA in the absence of RBM-B. In light of this, and the new data presented, we favour a model whereby PrimPol is recruited to chromatin primarily through the interaction between RBM-A and RPA70N. Once bound to chromatin, RBM-B is then able to bind a second RPA molecule, potentially further stabilising PrimPol on the DNA in order to facilitate repriming (we have now highlighted this possibility in the discussion). The importance of RBM-A over RBM-B may simply be due to the arrangement of the proteins on the DNA. RBM-A presumably binds the RPA molecule immediately adjacent to PrimPol, whereas RBM-B likely binds a second RPA molecule upstream of the first. This would make binding of RBM-B dependent on initial binding of RBM-A, which is consistent with our data. It is also possible the post-translational modification of either RPA or PrimPol, or both, plays a role in dictating the preference for recruitment via RBM-A. An alternative explanation, although one we do not have any data to support, would be that RBM-B binds to a different interacting partner in vivo, thus blocking access for binding to RPA.

2) The GFC analysis of the WT RBD domain shows an unusual early elution for such a small protein. They present dynamic light scattering and circular dichroism data to indicate that the protein is monomeric in solution. Yet, it would be more convincing if they could perform a SEC-MALS (Size Exclusion Chromatography with Multi-Angle Light Scattering) experiment that will provide an accurate measure for the molecular weight.

The unusually early elution for a small protein is a common problem when the protein is not globular. This arises because non-globular proteins run different through the matrix than globular proteins and the GFC calibration

curve is based on standards that are globular proteins. The DLS and CD fully support this explanation. SEC-MALS can provide another confirmatory measurement, but for non-globular proteins such as the RBD, it too is limited by intrinsic assumptions about the globularity of the proteins. Nevertheless, we have performed SEC-MALS experiments as requested to further confirm our analysis, and as expected, these new data indicate the protein is monomeric and non-globular. (described below in more detail)

3) When comparing the GFC from wild type RBD-RPA70N complex with the mutant versions RBD-A K-O and RBD-B K-O, it appears that the wild type RBD-RPA70N complex elutes at ~10.2ml, while the mutant version elute closer to 11ml. Is it possible that two RPA70N molecules can bind to one wild type RBD? A SEC-MALS experiment could easily resolve this (see also above).

We are in complete agreement with the reviewer that the WT RBD binds two RPA70N molecules, whereas the RBD-A-KO and RBD-B-KO each bind only one RPA70N molecule. We have performed additional SEC-MALS experiments as requested (described below in more detail). Within the limitations of above-noted caveats, the small difference in GFC elute volume observed in these experiments are fully consistent with this model.

4) In Fig 4 & 5, they show that the RBD-A motif is required for interaction with RPA and replication restart after DNA damage. Are the RBD-AK.O.(D519R/F522A) cells also more sensitive to UV light and/or DNA damaging agents?

We tested the sensitivity of RBM-A K.O. DT40 cells to UV damage using colony survival assays, however we do not see a significant decrease in survival in comparison to wild-type cells. This is not surprising given the relatively mild, although still significant, effect on replication fork speeds observed in these cells after UV damage. Importantly, the RBM-A K.O. mutant version of PrimPol possesses the same catalytic activity as the wild-type enzyme, mutation of RBM-A only affects the interaction with RPA and thus recruitment. However, given the level of overexpression in these cells, we expect that some PrimPol would still be recruited, simply due to the high levels of the enzyme present in the cell. Therefore, we would expect to see slowed fork speeds as binding of PrimPol to the DNA would undoubtedly be impaired in the absence of a functional RPA interaction for recruitment. However, given that PrimPol can bind ssDNA alone, it would likely present as a delay to fork restart rather than a complete inability. Indeed, we see that fork speeds are reduced after UV damage in these cells, but not to the same extent as in PrimPol-K.O. cells. This would explain why slowed forks are seen in fibre analysis, but no increase in UV sensitivity is observed.

Minor comments

5) Page 5, line4: "...interacts directly with RPA70N". Please indicate the boundaries of the construct (i.e. residues 1-120) .

6) Page 6 third section: RPA70NE7R. Please state in the text if the E7R mutation is close to, or distal from the RBD binding site, and if the mutation could affect the binding to RBD-A.

7) Page 7, second section: "PrimPol-RBD" and "PrimPol480-560" are both used to indicate the same construct. Please use only one (and indicate residues numbers if using PrimPolRBD).

All now corrected.

8) Page 8, first section, line 4. Change "70N" to "RPA70N"

Corrected

9) Page 8, second section, last 2 lines. "These interactions are of paramount importance in the binding of PrimPol to RPA70N". This is an odd statement given that later on in the manuscript it is shown that the RBD-B motif is dispensable for PrimPol-RPA interaction in vivo. Please change or remove.

To clarify that we are talking about the importance of the interactions in binding to RPA70N in vitro, we have corrected the sentence to "These electrostatic interactions are of paramount importance in the binding of PrimPol's RBM-B to RPA70N in vitro".

Reviewer #2:

The paper by Guilliam and colleagues builds on previous work by the same group and others (Guilliam et al, NAR, 2014; Wan et al, EMBO Rep, 2013), which showed that PrimPol recruitment to replication forks depends on the ssDNA-binding protein RPA, mediated by PrimPol's C-terminal domain. The novelty of this work lies in the identification of two RPA-binding motifs in PrimPol's C-terminal domain, the biophysical characterisation of the interaction and cell-based evidence of the physiological relevance of the interaction. The experiments appear to have been correctly executed and the experimental evidence presented is of good quality. The work addresses a narrow experimental point, and as such it will be of direct interest predominantly to PrimPol biologists.

Specific points:

If PrimPolRBD contains two RPA-binding domains of equal affinity, a PrimPol-RPA complex of 1:2 stoichiometry should be observable, in the GFC experiments or by other means.

As noted above for Reviewer 1, we have performed additional SEC-MALS experiment and these show PrimPol-RPA complex with 1:2 stoichiometry. (described below in more detail)

"Similar to PrimPolRBD, PrimPolA-K.O. eluted as a single defined multimeric complex from GFC (Fig. 3a)." What is the multimeric complex referred to here? The data in SupplFig 1 show that PrimPolRBD is monomeric in solution.

This sentence has been corrected to "Similar to results observed with PrimPol_{RBD}, PrimPol_{A-K.O.} and RPA70N eluted together as a single defined multimeric complex from GFC"

"Continuous electron density covers the entirety of RPA70NE7R and of the 15 residues in the synthesised PrimPol514-528 peptide, 9 residues (PrimPol517-525) are visible in the electron density maps. Within this short peptide, residues aspartate 516 to glutamate 524 are α -helical in content." Only residues 517-525 are visible in the map, but aspartate 516 is helical? Please clarify.

Thank you for pointing out this error, to the passage has been changed to "Within this short peptide, residues aspartate 519 to leucine 523 are α -helical in content."

The authors should report the PDB IDs for deposition of crystallographic models and structure factors.

We have now deposited these to the PDB.

Overview of SEC-MALS studies

SEC-MALS data was collected to address the reviewer comments prompted by the unexpected elution volume of RBD by analytical gel filtration. The focus of this study was to reconcile two competing results – an elution volume matching a molecular weight near tetramer by using a standard curve vs. DLS and NMR data suggestive of a monomer. A second topic raised by the reviewers concerns the stoichiometry of the complex and whether two RPA70N molecules can bind to one WT RBD molecule, suggesting this complex should be observable by SEC-MALS.

Methods

SEC-MALS was performed on an AKTA Purifier FPLC system (GE) connected to an Agilent Technologies RID and a Wyatt Technologies Dawn Helios 8+ MALS unit. A Superdex 75 increase 10/300 GL (24 mL) column was connected and equilibrated in running buffer consisting of 20 mM HEPES (pH 7.1), 80 mM NaCl, 0.5 mM TCEP. These conditions were chosen to mimic the buffer used for NMR titrations, and are similar to conditions used for other experiments. The flow was maintained at a consistent 0.5 mL/min and sample injections of 100 μ L from a static loop were initiated at the 0 mL point of each run. UV, RI, QELS, and LS values were recorded using ASTRA 6.1 (Wyatt). Estimated molecular weights for RBD and its saturated complexes were calculated using the Zimm algorithm surrounding the peak maximum and fit to a 0-order polynomial to give a single numerical estimate.

Data were collected using samples of RBD, RBD-A-KO, and RPA70N E7R at various ratios. The translated molecular weight of RPA70N is \sim 13.2 kDa and RBD is \sim 8.8 kDa. Concentrations for each sample were obtained after thawing and overnight dialysis in running buffer using UV 280nm absorbance and extinction coefficients calculated from the total number of tyrosine and tryptophan residues. RBD was run at concentrations of 185 μ M and RPA70N was added in 0, 1, 2 and 4x molar ratios. In order to compensate for high dilutions at 4:1, the baseline concentration of the sample was half of the others (92.5 μ M), and the absorbance normalized to the others after acquisition. RBD-A-KO was run at a calculated 185 μ M with and without 370 μ M RPA70N added.

Results

An injection of 100 μ L of 185 μ M RBD was run over the column to observe elution volume and collect MALS data sufficient for a molecular weight estimate. The signal from UV and RI detectors were normal, but the LS values were unexpectedly noisy.

After performing 5 runs with concentrations ranging from 185-400 μM , the run with the best baseline and most symmetrical LS peak was chosen for modelling (one of the 185 μM runs). The molecular weight was modelled from these data (Figure 1), giving an estimate that is similar to a monomer. The estimated weight is relatively flat at ~ 6 kDa near the leading edge of the peak, while the lagging edge of the peak slightly increases to 10-12 kDa. The 6 kDa estimate is slightly lower than the expected weight, but matches the DLS data that also gave a 6 kDa estimate. Regardless, the protein elutes as a monomer.

After referencing the DLS data, it is not surprising that the light scattering trace is noisy. According to DLS, the particle radius is ~ 1.25 nm. This is significantly lower than the specifications for the MALS instrument, which recommends a particle radius of ~ 10 nm or more. This was not a problem for the DLS instrument, which has specifications reaching 0.5 nm. However, using concentrations much higher than the MALS lower limit, we were able to overcome this size limitation and collect data with sufficient sensitivity for size modelling.

In order to address the issue of stoichiometry of RBD WT vs A-KO with saturating RPA70N, we performed a titration of RBD with RPA70N. We overlaid UV absorbance chromatograms to observe changes in the peak elution point and shape over the course of the titration (Figure 2). We observed that RBD eluted much earlier than RPA70N despite RBD having a lower molecular weight because it is non-globular. With one equivalent of RPA70N added, a bimodal peak appears with broadened

densities between a position near free RBD and a peak presumably of the complex (light grey trace). With two equivalents of RPA70N added, the peak at the RBD position is much weaker, while the complex elutes slightly earlier and increases in intensity (dark grey trace). With four equivalents of RPA70N added, the complex peak increases in intensity, the free RBD peak disappears, and a peak at the free RPA70N position becomes visible (black trace). Overall, this is indicative of a heterogeneous interaction, as would be expected from two binding sites of similar affinity. In addition, RBD can bind two RPA70N molecules simultaneously since the RPA70N peak is not present in the 2:1 trace, which addresses the reviewers comment. Figure 2 has been added to the supplementary figures (Figure S1a) and the data interpretation has been included in the main body of text to emphasize these findings.

To further verify the molecular weight of the saturated complex, MALS data were recorded for each trace, as well as for RBD-A-KO with 2x excess of RPA70N. The molecular weight estimate was performed for the 4:1 complex because it was the most saturated peak in the series. Because of the larger complex with RPA70N present, the LS values did not suffer from noise issues as observed previously. Unlike RBD by itself, the modeled molecular weight estimate is not a flat value across the peak (Figure 3). It is a flat value around ~38 kDa near the leading edge, which drops to ~20 kDa near the lagging edge. Our interpretation of this result is that the peak contains a heterogeneous mixture of singly and doubly bound RBD. For reference, one would expect a 1:1 complex to be ~22 kDa and the 2:1 complex to be ~36 kDa.

SEC-MALS was conducted on RBD-A-KO with 2-fold molar excess of RPA70N using the same procedure as WT RBD. Consistent with previous results, we observed RBD-A-KO eluting at a much later time than WT RBD. Modelling of this peak gave a predominantly flat MW estimate of ~22 kDa, very close to the expected value. This appears to confirm that this complex elutes stoichiometrically, unlike WT RBD. We note that there is a secondary species present at ~11-12 mL, but we do not know

what this is. It is likely a contaminant species in the original RBD-A-KO sample. The presence of this species does not alter the conclusion made from these data.

Conclusions

These data show that RBD is a monomer in solution, despite its unusual position in the elution profile. This is consistent with the DLS and NMR data. This discrepancy stems from the unusual protein shape that is the consequence of RBD being a non-globular protein (from CD and Xtal). Aberrant elution in gel filtration is well-known for such proteins.

SEC-MALS titration of RBD with RPA70N shows the existence of a 2:1 complex, but the elution profiles show a high degree of heterogeneity. This is confirmed by the non-linearity of the molecular weight estimate by MALS. Given this heterogeneity, it is remarkable how symmetrical the peak looked in our original 1:1 analytical gel filtration (original Figure S1a). We hypothesize this dissimilarity could be due to differences in the column, buffer pH, load volume, or load concentration.

MALS of RBM-A-KO shows that this heterogeneity is a function of stoichiometry and not RBD oligomerization. A clean 1:1 stoichiometry is expected here and is unambiguously observed.

The results presented by MALS are consistent with the DLS, NMR, and ITC data shown in the paper. It explains the extreme broadening by NMR upon titration (Fig S1d). The ITC data we show in the paper only uses RBM-A-KO and RBM-B-KO constructs. The WT RBD titration by ITC was very difficult to model and looked like multiple binding events, which is why it was not included in the paper.

REVIEWERS' COMMENTS:

Reviewer #1 (Remarks to the Author):

The authors have addressed all my comments, and I have no further questions.

Reviewer #2 (Remarks to the Author):

Nothing to comment.